# CViT: Continuous Vision Transformer for Operator Learning

**Sifan Wang**[1], **Jacob H. Seidman**[3,4,5], **Shyam Sankaran**[3], **Hanwen Wang**[2], **George J. Pappas**[4]
**Paris Perdikaris**[3]

[1]Institution for Foundation of Data Science, Yale University
[2]Graduate Program in Applied Mathematics and Computational Science, University of Pennsylvania
[3]Department of Mechanical Engineering and Applied Mechanics, University of Pennsylvania
[4]Department of Electrical and Systems Engineering, University of Pennsylvania
[5]Reality Defender

`sifan.wang@yale.edu`                          `{seidj,wangh19}@sas.upenn.edu`
`{shyamss,pappasg,pgp}@seas.upenn.edu`

## Abstract

Operator learning, which aims to approximate maps between infinite-dimensional function spaces, is an important area in scientific machine learning with applications across various physical domains. Here we introduce the Continuous Vision Transformer (CViT), a novel neural operator architecture that leverages advances in computer vision to address challenges in learning complex physical systems. CViT combines a vision transformer encoder, a novel grid-based coordinate embedding, and a query-wise cross-attention mechanism to effectively capture multi-scale dependencies. This design allows for flexible output representations and consistent evaluation at arbitrary resolutions. We demonstrate CViT's effectiveness across a diverse range of partial differential equation (PDE) systems, including fluid dynamics, climate modeling, and reaction-diffusion processes. Our comprehensive experiments show that CViT achieves state-of-the-art performance on multiple benchmarks, often surpassing larger foundation models, even without extensive pre-training and roll-out fine-tuning. Taken together, CViT exhibits robust handling of discontinuous solutions, multi-scale features, and intricate spatio-temporal dynamics. Our contributions can be viewed as a significant step towards adapting advanced computer vision architectures for building more flexible and accurate machine learning models in the physical sciences. All data and code are publicly available at `https://github.com/PredictiveIntelligenceLab/cvit`.

## 1 Introduction

Neural operators (Kovachki et al., 2023) have emerged as a powerful class of deep learning models for scientific machine learning applications, particularly in building efficient surrogates for expensive partial differential equation (PDE) solvers. These models aim to learn mappings between infinite-dimensional function spaces, enabling rapid acceleration of complex simulations in fields such as fluid dynamics and climate modeling (Azizzadenesheli et al., 2024; Serrano et al., 2023). Popular architectures like Fourier Neural Operators (FNO) (Li et al., 2021; Tran et al., 2021) and DeepONet (Lu et al., 2021; Wang et al., 2022) have shown promising results, supported by universal approximation guarantees (Chen & Chen, 1995; Lanthaler et al., 2022; Kovachki et al., 2021; De Ryck & Mishra, 2022) and interpretations as learning low-dimensional manifolds in function spaces (Seidman et al., 2022; 2023).

However, the architectural design of current neural operators has largely been motivated by mathematical intuition about the structure of PDE solution spaces, rather than leveraging recent advances in deep learning architectures. This approach, while theoretically sound, may limit the performance and flexibility of these models when applied to complex, real-world physical systems.

In parallel, the field of computer vision has seen significant advancements in neural field models, which represent continuous functions parameterized by neural networks (Stanley, 2007). A key innovation in this domain has been the development of conditioned neural fields, which allow for auxiliary inputs to modulate the field's behavior (Xie et al., 2022). These models have demonstrated remarkable effectiveness in tasks such as representing high-resolution images and 3D scenes (Mildenhall et al., 2021; Müller et al., 2022).

Motivated by the success of conditioned neural fields in computer vision, we introduce the Continuous Vision Transformer (CViT), a novel neural operator architecture that bridges the gap between advanced computer vision techniques and scientific machine learning. Our main contributions can be summarized as follows:

- **Continuous Vision Transformer (CViT).** We introduce CViT, a neural operator that combines a vision transformer encoder with a novel, trainable grid-based coordinate embedding in its decoder. This design enables CViT to effectively capture multi-scale spatial dependencies and allows for continuous querying at arbitrary resolutions.
- **Theoretical Insights into Lipschitz Constant and Spectral Bias.** We present novel insights into the impact of coordinate embeddings on the Lipschitz constant of neural fields and neural operators. This analysis supports the choice of the proposed grid-based coordinate embedding and its efficacy in mitigating spectral bias, a common limitation in training neural operators .
- **State-of-the-art Performance.** Despite its architectural simplicity, CViT achieves state-of-the-art results on challenging benchmarks in climate modeling, fluid dynamics and reaction-diffusion processes. It often outperforms larger models while using fewer parameters and without requiring extensive pretraining.

Taken together, we explore the connection between operator learning architectures and conditioned neural fields, offering a unifying perspective for examining differences among popular operator learning models (see Appendices B and C). This perspective not only enhances understanding of existing architectures but also provides a general framework for designing future neural operators. By bridging the gap between computer vision and scientific machine learning, CViT paves the way for improved surrogates of complex physical systems, with potential applications ranging from climate modeling to engineering design.

## 2  BACKGROUND AND RELATED WORK

**Background.**  The emergence of neural operators has driven advancements in solving partial differential equations (PDEs) and modeling physical systems. Building upon the foundational architectures mentioned in the introduction, recent developments have focused on enhancing the expressiveness and efficiency of these models. For instance, extensions of the Fourier Neural Operator (FNO) (Li et al., 2021) have explored factorized representations (Tran et al., 2021) to reduce computational complexity while maintaining performance. Similarly, wavelet-based approaches (Gupta et al., 2021; Tripura & Chakraborty, 2022) have been proposed to capture multi-scale features more effectively. Most existing neural operator architectures can be interpreted as conditioned neural fields (Xie et al., 2022) with specific choices of base fields and conditioning mechanisms, see Appendices B and C for a detailed discussion.

**Transformer-Based Operator Learning.**  Concurrently, the success of vision transformers in computer vision tasks has inspired their adaptation to scientific machine learning problems. The self-attention mechanism central to transformers offers a powerful tool for capturing long-range dependencies in spatial and temporal data, a crucial aspect in many PDE-governed systems. For example, OFormer (Li et al., 2022) introduced a novel way to embed continuous input functions and output queries into a transformer architecture. This approach demonstrated the potential of transformers in handling the inherent continuity of physical problems. GNOT (Hao et al., 2023) further extended this idea by incorporating graph structures, allowing for more flexible representation of input functions and improved handling of irregular domains. A significant advancement in this direction came with DPOT (Hao et al., 2024), which introduced a denoising pre-training strategy. This approach enables the model to learn from diverse PDE datasets, significantly enhancing its generalization capabilities.

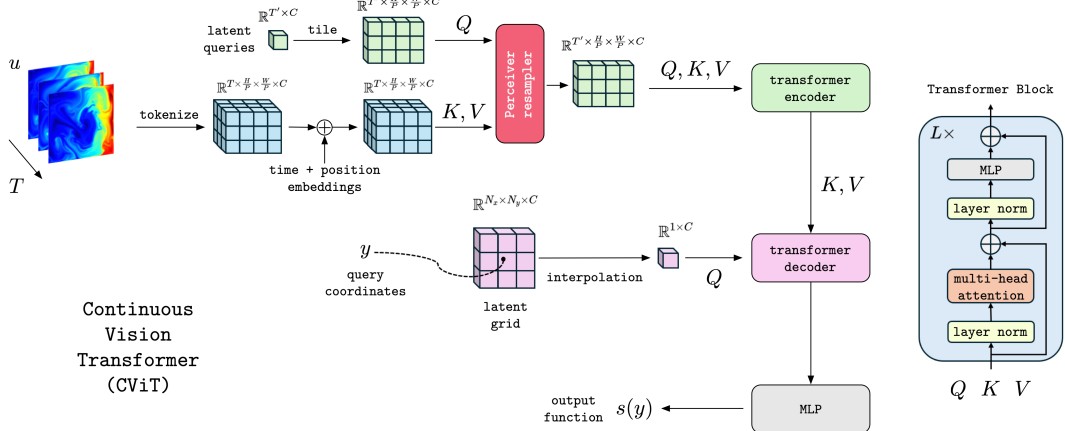

Figure 1: *Continuous Vision Transformer (CViT) Architecture:* CViT consists of the following components: (1) Spatio-temporal patch embeddings to extract localized features. (2) A temporal aggregation module based on the Perceiver architecture, which captures temporal correlations to compresses tokens along the time axis. (3) A Transformer encoder that captures multi-scale spatial dependencies via self-attention layers. (4) A novel grid-based positional encoding scheme for query coordinates, allowing for flexible output representation and interpolation. (5) A cross-attention decoder that integrates information from the input function with query coordinates.

**Open Challenges.** Despite these advancements, several key challenges remain in the field of operator learning, particularly in addressing resolution independence, high-dimensional data, long-range spatio-temporal dependencies and trade-offs between model expressiveness and computational efficiency. Transformers typically rely on fixed-resolution data and positional encodings, which hinder their ability to generalize across varying scales or irregular domains commonly encountered in scientific problems. Efficiently capturing global and local dependencies in high-dimensional systems further compounds computational demands, especially as self-attention scales quadratically with input size. These constraints make it difficult to balance scalability, precision, and efficiency. Addressing even a subset of these challenges requires novel architectural designs that can leverage the strengths of both neural operators and vision transformers while overcoming their respective limitations. This sets the stage for our proposed Continuous Vision Transformer (CViT), which aims to tackle specific challenges related to resolution independence and efficient capturing of multi-scale features in regular grid-based problems, combining the continuous nature of neural operators with the powerful representation capabilities of vision transformers.

## 3 CONTINUOUS VISION TRANSFORMER (CVIT)

While neural operators have shown promising results in learning mappings between function spaces, their architectural design has not fully leveraged recent advancements in deep learning. Popular architectures like FNO (Li et al., 2021), DeepONet (Lu et al., 2021), and NoMaD (Seidman et al., 2022) (see Appendix C) primarily rely on simple building blocks such as Fourier layers or fully connected networks, which may not effectively capture complex spatial dependencies and multiscale features present in many physical systems.

Motivated by the success of vision transformers in computer vision tasks, we introduce the Continuous Vision Transformer (CViT), a novel neural operator architecture that combines the strengths of vision transformers and continuous function representations. CViT aims to learn more powerful and flexible representations of time-dependent physical systems by leveraging a vision transformer encoder to capture complex spatial dependencies in input functions, a trainable grid-based positional encoding for flexible representation of query coordinates, and a cross-attention mechanism to integrate input function information with query coordinates.

Taken together, the overall design of CViT's architecture draws inspiration from the analysis of existing neural operators presented in Appendix C, while leveraging the power of vision transformers to enhance expressiveness.

## 3.1 Architecture Description

The CViT architecture consists of three main components: a vision transformer encoder, a grid-based positional encoding, and a cross-attention mechanism.

**Patch Embeddings.** The vision transformer encoder takes as input a gridded representation of the input function $u$, yielding a spatio-temporal data tensor $\mathbf{u} \in \mathbb{R}^{T \times H \times W \times D}$ with $D$ channels. We patchify our inputs into 3D tokens $\mathbf{u}_p \in \mathbb{R}^{T \times \frac{H}{P} \times \frac{W}{P} \times C}$ by tokenizing each 2D spatial frame independently, following the process used in standard Vision Transformers (Dosovitskiy et al., 2020). We then add trainable 1D temporal and 2D spatial positional embeddings to each token:

$$\mathbf{u}_{pe} = \mathbf{u}_p + \mathrm{PE}_t + \mathrm{PE}_s, \quad \mathrm{PE}_t \in \mathbb{R}^{T \times 1 \times 1 \times C}, \quad \mathrm{PE}_s \in \mathbb{R}^{1 \times \frac{H}{P} \times \frac{W}{P} \times C}. \tag{1}$$

**Temporal Aggregation.** To reduce computational cost, we use a temporal aggregation layer based on the Perceiver architecture (Jaegle et al., 2021; Alayrac et al., 2022). Specifically, we learn a pre-defined number of latent input queries $\mathbf{z} \in \mathbb{R}^{T' \times C}$. These queries serve as learnable parameters in a cross-attention module, which processes the visual features of our input. The Perceiver module operates on flattened visual features $\mathbf{u}_f \in \mathbb{R}^{(\frac{H}{P} \times \frac{W}{P}) \times T \times C}$ obtained by flattening the positionally encoded features $u_{pe}$. Through this cross-attention mechanism, we aggregate the temporal information into a compact latent representation $\mathbf{z}_{agg} \in \mathbb{R}^{(\frac{H}{P} \times \frac{W}{P}) \times T' \times C}$ as:

$$\mathbf{z}' = \hat{\mathbf{z}} + \mathrm{MHA}\left(\mathrm{LN}\left(\hat{\mathbf{z}}\right), \mathrm{LN}\left(\mathbf{u}_f\right), \mathrm{LN}\left(\mathbf{u}_f\right)\right), \tag{2}$$

$$\mathbf{z}_{agg} = \mathbf{z}' + \mathrm{MLP}\left(\mathrm{LN}\left(\mathbf{z}'\right)\right). \tag{3}$$

Here, $\mathbf{z}$ is initialized by a unit Gaussian distribution and $\hat{\mathbf{z}} \in \mathbb{R}^{(\frac{H}{P} \times \frac{W}{P}) \times T' \times C}$ is obtained by tiling the latent query $\mathbf{z}$ $\left(\frac{H}{P} \times \frac{W}{P}\right)$ times. Besides compressing tokens over the time axis, this module also enables the model to handle inputs with a variable number of time steps.

**Transformer Encoder.** We then process the aggregated tokens $\mathbf{z}_{agg}$ using a sequence of $L$ pre-norm Transformer blocks (Vaswani et al., 2017; Xiong et al., 2020),

$$\mathbf{z}_0 = \mathrm{LN}(\mathbf{z}_{agg}), \tag{4}$$

$$\mathbf{z}'_\ell = \mathrm{MSA}\left(\mathrm{LN}\left(\mathbf{z}_{\ell-1}\right)\right) + \mathbf{z}_{\ell-1}, \qquad \ell = 1 \dots L, \tag{5}$$

$$\mathbf{z}_\ell, = \mathrm{MLP}\left(\mathrm{LN}\left(\mathbf{z}'_\ell\right)\right) + \mathbf{z}'_\ell, \qquad \ell = 1 \dots L. \tag{6}$$

**Cross-Attention Decoder.** To enable continuous evaluation of outputs, we design a novel and efficient coordinate embedding to capture the fine-scale features of the target functions. Specifically, we create a uniform grid $\{\mathbf{y}_{ij}\} \subset [0,1]^2$, for $i = 1, \dots N_x$ and $j = 1, \dots N_y$, along with associated trainable latent grid features $\mathbf{x} \in \mathbb{R}^{N_x \times N_y \times C}$. For a query point $y \in \mathbb{R}^2$, we compute a Nadaraya-Watson interpolant over grid latent features:

$$\mathbf{x}' = \sum_{i=1}^{N_x} \sum_{j=1}^{N_y} w_{ij} \mathbf{x}_{ij}, \quad w_{ij} = \frac{\exp\left(-\beta \|y - \mathbf{y}_{ij}\|^2\right)}{\sum_{ij} \exp\left(-\beta \|y - \mathbf{y}_{ij}\|^2\right)}. \tag{7}$$

Here $\beta > 0$ is a hyperparameter that determines the locality of the interpolated features. It is important for determining the smoothness of the interpolation function. Specifically, larger values of $\beta$ yield more localized weight distributions $w_{ij}$, resulting in a higher-frequency interpolant that captures finer-scale variations. Conversely, smaller $\beta$ values produce a smoother interpolant by averaging over a broader neighborhood of points.

The interpolated grid feature $\mathbf{x}_0 = \mathbf{x}' \in \mathbb{R}^{1 \times C}$ is used as query input to a transformer decoder. This decoder uses the output of the vision transformer encoder $\mathbf{z}_L$ as keys and values in a cross-attention mechanism:

$$\mathbf{x}'_k = \mathbf{x}_{k-1} + \mathrm{MHA}\left(\mathrm{LN}\left(\mathbf{x}_{k-1}\right), \mathrm{LN}\left(\mathbf{z}_L\right), \mathrm{LN}\left(\mathbf{z}_L\right)\right), \qquad k = 1 \dots K, \tag{8}$$

$$\mathbf{x}_k = \mathbf{x}'_k + \mathrm{MLP}\left(\mathrm{LN}\left(\mathbf{x}'_k\right)\right), \qquad k = 1 \dots K. \tag{9}$$

Finally, a small MLP network projects the output to the desired output dimension.

**Theoretical Insights.** The theoretical analysis of Lipschitz constants for different coordinate embeddings (as detailed in Appendix E) provides crucial insights that inform the design choices in CViT and have broader implications for neural operator architectures. Our findings reveal that the choice of coordinate embedding significantly influences the decoder's capacity to learn complex, high-frequency functions.

For conventional MLP decoders with linear coordinate embeddings, the Lipschitz constant is approximately 1 at initialization, which does not enhance the network's ability to capture fine-scale features, thereby leading to prediction susceptible to spectral bias (Rahaman et al., 2019). In contrast, random Fourier features can increase the Lipschitz constant through two mechanisms: by increasing the standard deviation of the sampling distribution or by increasing the network width. This observation not only explains the effectiveness of Fourier features in addressing spectral bias, but also highlights a potential trade-off between network size and the frequency range that can be captured.

Our proposed grid-based coordinate embedding in CViT offers a more direct and controllable approach to increasing the network's Lipschitz constant. By adjusting the interpolation parameter $\beta$, we can tune the model's ability to capture high-frequency components in the target functions. This provides a clear advantage as it allows for more precise control over the model's spectral properties without necessarily increasing model complexity.

**CViT vs other Transformer-based Approaches.** To further highlight the key innovations of CViT, we compare it with other popular transformer-based operator learning methods:

*OFormer* (Li et al., 2022): OFormer uses an encoder-decoder structure. Its encoder combines a point-wise MLP and stacked self-attention blocks to process input functions, while employing random Fourier Features (Tancik et al., 2020) for query coordinate encoding. The model generates latent embeddings through cross-attention at sampled query locations. In the decoder, OFormer uses recurrent MLPs for dynamic propagation instead of traditional masked attention. Unlike CViT, it doesn't use ViT's patch embedding to tokenize input functions, potentially limiting its ability to capture complex spatial dependencies.

*GNOT* (Hao et al., 2023): GNOT is a transformer-based framework for learning operators that addresses the challenges of multiple input functions and irregular meshes. It designs a heterogeneous normalized attention layer, providing an encoding interface for various input functions and additional prior information. Additionally, GNOT features a geometric gating mechanism, described as a soft domain decomposition approach to multi-scale problems. Unlike CViT, GNOT uses an plain MLP to process query coordinates, which may potentially suffer from spectral bias (Rahaman et al., 2019).

*DPOT* (Hao et al., 2024): DPOT is a transformer-based architecture designed for pretraining on multiple PDEs. It introduces a new auto-regressive denoising strategy and a novel model architecture based on Fourier attention. Unlike OFormer, GNOT, and CViT, DPOT does not explicitly take query coordinates as inputs, which implies that it cannot be evaluated at arbitrary locations.

*MPP* (McCabe et al., 2023): MPP is a large-scale PDE surrogate model that directly employs a Vision Transformer as its backbone. Similar to DPOT, it does not take query coordinates as inputs, thereby losing flexibility in diverse output representations.

## 4 EXPERIMENTS

We compare the CViT against popular neural operators on three challenging benchmarks in physical sciences. In addition to demonstrating CViT's performance against strong baselines, we also conduct comprehensive ablation studies to probe the sensitivity on its own hyper-parameters.

**CViT model setup.** We construct CViT models with different configurations, as summarized in Table 1. For all experiments, unless otherwise stated, we use a patch size of $8 \times 8$ for tokenizing inputs. We also employ a decoder with a single cross-attention Transformer block for all configurations. While we tested decoders with multiple layers, they did not yield improvements. The grid resolution is set to the spatial resolution of each dataset. The latent dimension of grid features is set to 512, and if it does not match the transformer's embedding dimension, we align it using a dense layer. Besides, we use $\beta = 10^5$ to ensure sufficient locality of the interpolated features. These choices are validated

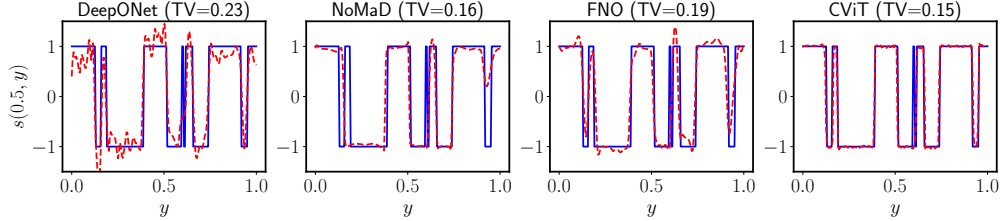

Figure 2: *Advection of discontinuous waveforms.* Prediction (red dashed line) versus ground truth (blue line) for the worst-case example in the test dataset, for: DeepONet; NoMaD; FNO; CViT. Also reported is the associated Total Variation (TV) error ($\downarrow$).

by extensive ablation studies, as illustrated in Figure 4. Full details of the training and evaluation procedures are provided in Appendix F.1.

Table 1: Details of Continuous Vision Transformer model variants.

| Model | Encoder layers | Embedding dim | MLP width | Heads | # Params |
|---|---|---|---|---|---|
| CViT-S | 5 | 384 | 384 | 6 | 13 M |
| CViT-B | 10 | 512 | 512 | 8 | 30 M |
| CViT-L | 15 | 768 | 1536 | 12 | 92 M |

**Baselines.** We select the following methods as our baselines for comparisons. **DeepONet** (Lu et al., 2021): One of the first neural operators with universal approximation guarantees. **NoMaD** (Seidman et al., 2022): A recently proposed architecture leveraging nonlinear decoders to achieve enhanced dimensionality reduction. **FNO** (Li et al., 2021): An efficient framework for operator learning in the frequency domain. **FFNO** (Tran et al., 2021): An improved version of FNO utilizing a separable Fourier representation, which reduces model complexity and facilitates deeper network structures. **UNO** (Ashiqur Rahman et al., 2022): A U-shaped neural operator with FNO layers. **U-Net$_{att}$** (Gupta & Brandstetter, 2022): A modern U-Net architecture using bias weights and group normalization, and wide ResNet-style 2D convolutional blocks, each of which is followed by a spatial attention block. **U-F2Net** (Gupta & Brandstetter, 2022): A U-Net variant where the lower blocks in both the downsampling and upsampling paths are replaced by Fourier blocks, each consisting of 2 FNO layers with residual connections. **Dilated ResNet** (Stachenfeld et al., 2021): A ResNet model that adapts filter sizes at different layers using dilated convolutions, providing an alternative method for aggregating global information. **GK-Transformer** (Cao, 2021) / **OFormer** (Li et al., 2022) / **GNOT** (Hao et al., 2023): Various transformer-based architectures for operator learning, with different designs of their encoder/decoder and attention mechanisms. **DPOT** (Hao et al., 2024) / **MPP** (McCabe et al., 2023): Transformer-based networks with patch embedding as in ViT, and pre-trained on extensive PDE datasets using a rollout loss.

For all baselines, we report evaluation results from the original papers when applicable. When not applicable, we train and evaluate each model following the suggested settings in the respective paper. Details on the implementation of baseline models are provided in Appendix F.2.

### 4.1 MAIN RESULTS

Here we provide our main results across three challenged benchmarks. The full details on the underlying equations, dataset generation and problem setup for each case are provided in the Appendix; see Section F.3, F.4 and F.5, respectively.

**Advection of discontinuous waveforms.** The first benchmark involves predicting the transport of discontinuous waveforms governed by a linear advection equation with periodic boundary conditions. We make use of the datasets and problem setup established by Hoop *et al.* (de Hoop et al., 2022). This benchmark evaluates our model's capability in handling discontinuous solutions and shocks in comparison to popular neural operator models.

Figure 2 illustrates the test sample corresponding to the worst-case prediction of each model we compared. We observe that CViT is able to better capture the discontinuous targets and yield the

most sharp predictions across all baselines. This is also reflected in both the relative L2 and Total Variation error metrics; see Appendix Table 5 for a more complete summary, including details on model configurations, as well as average, median and worst case errors across the entire test dataset.

**Shallow-water equations.** The second benchmark involves a fluid flow system governed by the 2D shallow-water equations. This describes the evolution of a 2D incompressible flow on the surface of the sphere and it is commonly used as a benchmark in climate modeling. Here we adhere to the dataset and problem setup established in PDEArena (Gupta & Brandstetter, 2022), allowing us to perform fair comparisons with several state-of-the-art deep learning surrogates.

Table 2 presents the results of CViT against several competitive and heavily optimized baselines. Our proposed method achieves the lowest relative $L^2$ error. It is also worth noting that the reported performance gains are achieved using CViT configurations with a significantly smaller parameter count compared to the other baselines. This highlights the merits of our approach that enables the design of parameter-efficient models whose outputs can also be continuously queried at any resolution. Additional visualizations of our models are shown in Figure 3 and 10.

Table 2: Performance for 5-step roll-out predictions in the shallow-water equations benchmark.

| Model | # Params | Rel. $L^2$ error ($\downarrow$) |
|---|---|---|
| DilResNet | 4.2 M | 13.20% |
| U-Net$_{att}$ | 148 M | 5.68% |
| FNO | 268 M | 3.97% |
| U-F2Net | 344 M | 1.89% |
| UNO | 440 M | 3.79% |
| **CViT-S** | 13 M | **4.47%** |
| **CViT-B** | 30 M | **2.69%** |
| **CViT-L** | 92 M | **1.56%** |

In addition, here we perform extensive ablation studies on the hyper-parameters of CViT. The results are summarized in Figure 4. As shown in Figure 4 (a), we observe that a smaller patch size typically leads to better accuracy, but at higher computational costs. Moreover, Figure 4 (b) demonstrates that the model performance is significantly influenced by the type of coordinate embeddings used in the CViT decoder. We compare our proposed grid-based embedding against a small MLP and random Fourier features (Tancik et al., 2020). The grid-based embedding achieves the best accuracy, outperforming other methods by up to an order of magnitude. This result is in agreement with the theoretical insights presented in Appendix E.

We also investigate the impact of the resolution of the associated grid. Figure 4 (c) reveals that the best results are obtained when the grid resolution matches the highest resolution at which the model is evaluated. As another direct empirical evidence of theoretical results in Appendix E, Figure 4(d) shows the CViT model's sensitivity to the hyper-parameter $\beta$ used for computing interpolated features; too small $\beta$ values can degrade predictive accuracy. Finally, in Appendix Figure 9, we observe minor improvements by increasing the latent dimension of grid features, or the number of decoder attention heads. Hence the overall model performance remains robust to variations in these hyper-parameters.

Taken together, these findings not only justify our design choices for CViT but also provide general insights for the development of neural operators. They suggest that future research in this field should pay close attention to the interplay between spatio-temporal tokenization, coordinate representation, and the inherent multi-scale nature of physical systems. Such considerations could lead to even more effective architectures for a wide range of scientific machine learning tasks.

**Navier-Stokes equations (NS).** Our third benchmark focuses on the compressible and incompressible Navier-Stokes equations, which underpin a variety of systems, ranging from hydromechanical processes to weather forecasting and turbulent dynamics analysis. For a fair comparison with various neural operators and other state-of-the-art deep learning models, we use the dataset generated by PDEArena (Gupta & Brandstetter, 2022), and PDEBench Takamoto et al. (2022). and follow the problem setup reported in Hao *et al.* (Hao et al., 2024).

Our main results are summarized in Table 3, indicating that CViT outperforms all baselines we have considered. Notably, the previous best result (DPOT-L (Fine-tuned)) is achieved with a much larger transformer-based architecture, which involves pretraining on massive diverse PDE datasets followed by fine-tuning on the target dataset for 500 epochs. Here we also note that DPOT was pretrained and fine-tuned using a rollout loss, which directly optimizes for the multi-step prediction task on which the models are evaluated. In contrast, CViT was trained using a one-step loss without any rollout

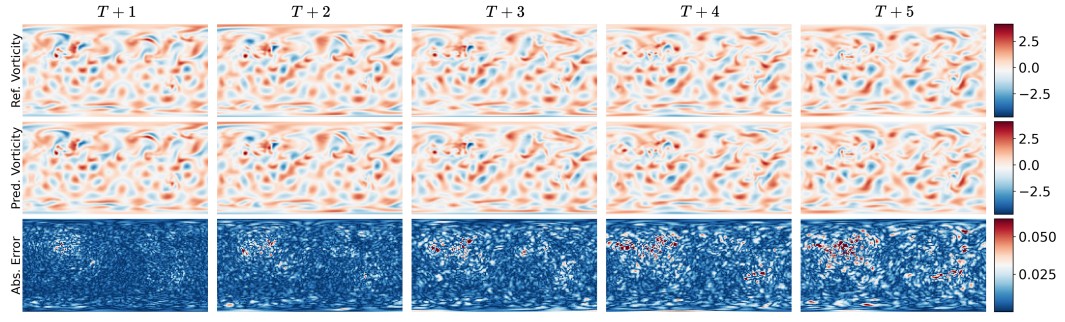

Figure 3: *Shallow water benchmark.* Representative CViT rollout prediction of the vorticity field, an...

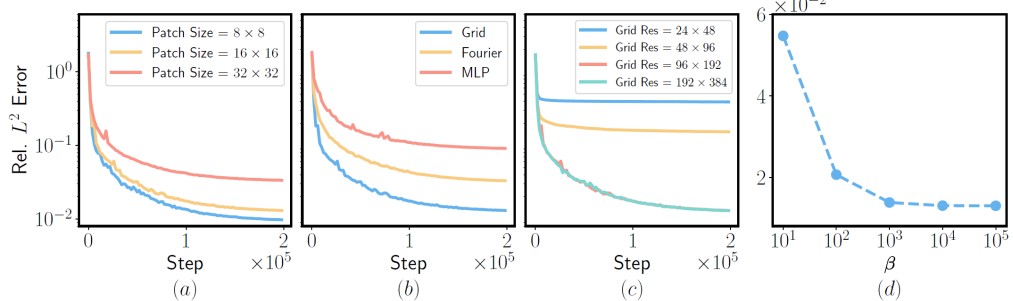

Figure 4: *Ablation studies for CViT on the shallow-water equations benchmark.* Convergence of test errors for: (a) different patch sizes; (b) different coordinate embeddings; (c) different resolutions of latent grids; (d) sensitivity on interpolating the CViT latent grid features, as controlled by the $\beta$ parameter. Results obtained using CViT-L with $16 \times 16$ patch-size, varying each hyper-parameter of interest while keeping others fixed.

finetuning. This difference in training objective significantly favors DPOT for rollout evaluations across all benchmarks. Furthermore, CViT exhibits remarkable parameter efficiency. CViT-S, with only 13M parameters, achieves accuracy comparable to the largest pretrained model (1.03B). Additionally, CViT shows good scalability; as the number of parameters increases, performance improves. These findings strongly support the effectiveness our approach in learning evolution operators for complex physical systems. Additional results and sample visualizations are provided in Figure 5, as well as Figure 13 and 14 in Appendix.

Table 3: Relative $L^2$ error of rollout predictions on the incompressible and compressible Navier-Stokes and Diffusion-Reaction benchmark.

| Model | # Params | NS | CNS | DR |
|---|---|---|---|---|
| FNO | 0.5 M | 9.12 % | 9.60 % | 12.00 % |
| FFNO | 1.3 M | 8.39 % | 5.20 % | 5.71 % |
| GK-T | 1.6 M | 9.52% | 3.77% | 3.59 % |
| GNOT | 1.8 M | 17.20 % | 4.20 % | 3.11 % |
| Oformer | 1.9 M | 13.50 % | 6.25 % | 1.92 % |
| DPOT-Ti | 7 M | 12.50 % | 3.97 % | 3.21 % |
| MPP-S | 30M | - | - | 1.12 % |
| DPOT-S | 30 M | 9.91 % | 3.37 % | 3.79 % |
| MPP-L (Pre-trained) | 400 M | - | - | 0.98 % |
| DPOT-L (Pre-trained) | 500 M | 7.98 % | 2.16 % | 2.32 % |
| DPOT-L (Fine-tuned) | 500 M | 2.78 % | 1.31 % | 0.73 % |
| DPOT-H (Pre-trained) | 1.03 B | 3.79 % | 1.80 % | 1.91 % |
| **CViT-S** | 13 M | **3.75 %** | **2.71 %** | **1.13 %** |
| **CViT-B** | 30 M | **3.18 %** | **1.99 %** | **1.11 %** |
| **CViT-L** | 92 M | **2.35 %** | **1.29 %** | **0.68 %** |

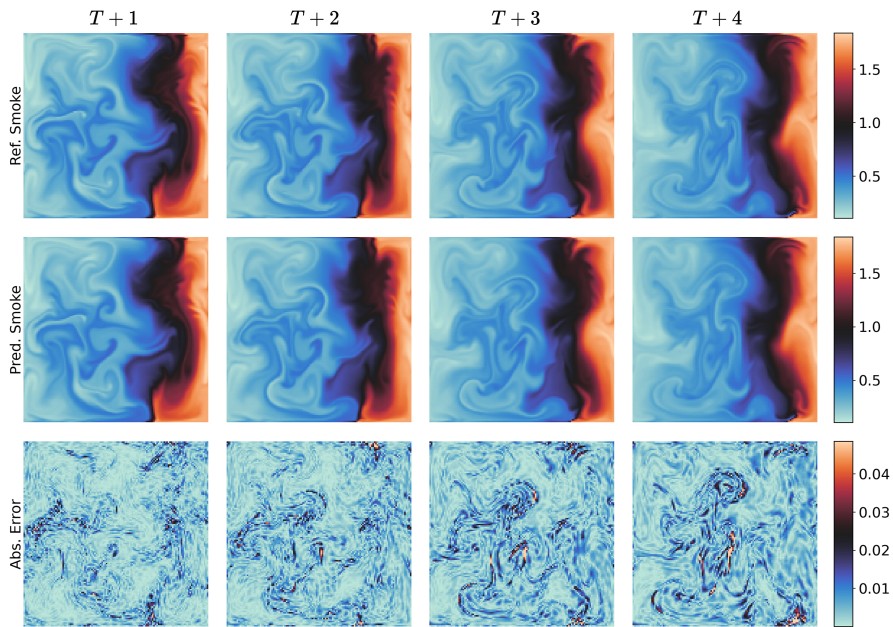

Figure 5: *Incompressible Navier-Stokes benchmark (NS)*. Representative CViT rollout predictions of the passive scalar field, and point-wise error against the ground truth.

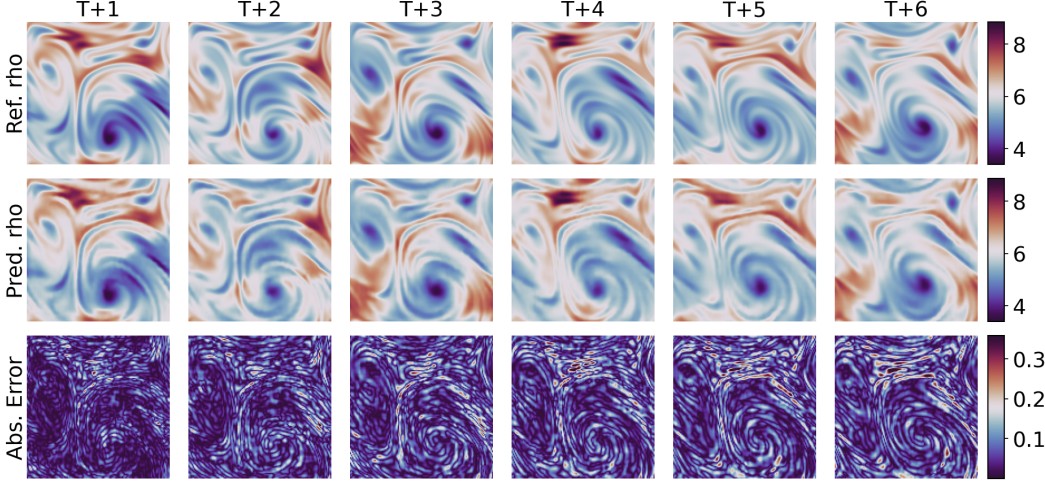

Figure 6: *Compressible Navier-Stokes Benchmark (CNS)*. Representative CViT rollout prediction of the density field $\rho$, and point-wise error against the ground truth.

**Diffusion-Reaction equations (DR).** Our final benchmark focuses on learning the density fields $\mathbf{u} = (u, v)$ governed by a coupled 2D diffusion-reaction equation (DR). This PDE, commonly used in chemistry, models substances reacting and diffusing across a spatial domain, illustrating chemical reaction processes and resulting in diverse spatial patterns. For a fair comparison, we use the dataset from PDEBench (Takamoto et al., 2022) and follow the problem setup described by Hao et al. (2024).

Table 3 presents our main results. Consistent with previous findings, CViT-L achieves the best performance across all baselines. CViT models outperform other approaches with comparable or fewer parameters, demonstrating good scalability and parameter efficiency. Representative rollout predictions are provided in Figure 18 and 19 (see Appendix). Our predictions show excellent agreement with the corresponding ground truth.

Interestingly, we note that the solution of Diffusion-Reaction equations appears simpler than that of the Navier-Stokes equation dataset, resulting in uniformly lower test errors for various baselines in the DR benchmark. In particular, OFormer outperforms GNOT in both examples, despite having

comparable parameters. We postulate that this performance gain is due to the use of random Fourier features in OFormer, facilitating more efficient learning of complex functions. This observation further demonstrates the importance of coordinate embedding in designing neural operators and indirectly highlights the effectiveness of our proposed grid-based embedding.

## 5 DISCUSSION

**Summary.**  This work introduces the Continuous Vision Transformer (CViT), a novel neural operator architecture that leverages advances in computer vision to address challenges in learning complex physical systems. CViT combines the strengths of vision transformers and continuous function representations, achieving state-of-the-art accuracy on challenging benchmarks in climate modeling, fluid dynamics and reaction-diffusion processes. Our approach demonstrates the potential of adapting advanced computer vision techniques to develop more flexible and accurate machine learning models for physical sciences. The key innovations of CViT include: (a) a vision transformer encoder for capturing multi-scale spatial dependencies in input functions, (b) a trainable grid-based positional encoding for flexible representation of query coordinates, and (c) a query-wise cross-attention mechanism ensuring consistent evaluation at arbitrary resolutions. These design choices ensure that CViT maintains the desirable properties of a well-defined conditioned neural field (as discussed in Appendix B), while offering improved flexibility and performance (as discussed in Appendix E).

Our empirical results across various PDE benchmarks demonstrate that CViT often outperforms larger models with fewer parameters and without extensive pretraining. This highlights the merits of our approach in designing parameter-efficient models that can be continuously queried at any resolution. The broader impact of this work lies in its potential to accelerate scientific discovery through more efficient and accurate simulations of complex physical systems, with applications ranging from climate modeling to engineering design.

**Limitations.**  Despite its promising performance, the proposed CViT architecture can be further improved in multiple areas. Firstly, like most transformer-based models, CViT's self-attention blocks scale quadratically with the number of tokens, posing computational challenges for high-dimensional or high-resolution data. Recent advancements in computer vision (Liu et al., 2021b) are tackling these scalability issues, and these ideas can be transferred to our setting to improve CViT's efficiency for larger-scale problems. Secondly, transformer-based models like CViT generally require large training datasets for optimal performance and may struggle in data-scarce scenarios without careful regularization. To address this, future work could explore leveraging CViT's ability to reconstruct continuous output functions in conjunction with physics-informed loss functions (Wang et al., 2021; 2023) for more data-efficient training and improved generalization. Lastly, while our current experiments treat input functions as sequences of images, many practical applications in scientific computing involve complex geometric structures and diverse input modalities such as unstructured meshes or point clouds. Addressing these cases requires developing appropriate tokenization schemes, and recent work in (Alkin et al., 2024; Pang et al., 2022) has taken steps in this direction. Extending CViT to handle such diverse input modalities represents an important avenue for future research.

**Future Work.**  Building on the success of CViT, several promising directions for future research emerge. One key area is the exploration of more efficient attention mechanisms to improve scalability to higher-dimensional problems. Additionally, investigating the potential of pre-training strategies, similar to those employed in DPOT (Hao et al., 2024), could further enhance CViT's performance and generalization capabilities across diverse PDE systems. Finally, extending CViT to handle 3D problems and developing techniques for multi-resolution training and evaluation could significantly broaden its applicability in scientific computing domains.

**Broader Impact.**  The broader impact of this work lies in its potential to fuel the development of resolution-invariant foundation models for accelerating scientific discovery and enable more efficient and accurate simulations of complex physical systems, with applications ranging from climate modeling to engineering design.

ACKNOWLEDGMENT

We would like to acknowledge support from the US Department of Energy under the Advanced Scientific Computing Research program (grant DE-SC0024563) and Yale Institute for Foundations of Data Science. We also thank the developers of the software that enabled our research, including JAX Bradbury et al. (2018), Matplotlib Hunter (2007), and NumPy Harris et al. (2020).

ETHICS STATEMENT

Our contributions enable advances in accelerating the modeling and emulation of complex physical systems. This has the potential to significantly impact fields such as climate science, fluid dynamics, and materials engineering by providing more accurate and efficient simulation tools. While we do not anticipate specific negative impacts from this work, as with any powerful predictive tool, there is potential for misuse. We encourage the research community to consider the ethical implications and potential dual-use scenarios when applying these technologies in sensitive domains.

REPRODUCIBILITY STATEMENT

The code used to carry out the ablation experiments is provided as a supplementary material for this submission. All data and code are publicly available at `https://github.com/PredictiveIntelligenceLab/cvit`.

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

## A   NOMENCLATURE

Table 4 summarizes the main symbols and notation used in this work.

Table 4: Summary of the main symbols and notation used in this work.

| Notation | Description |
|---|---|
| **Operator Learning** | |
| $\mathcal{X}$ | The input function space |
| $\mathcal{Y}$ | The output function space |
| $u \in \mathcal{X}$ | Input function |
| $s \in \mathcal{Y}$ | Output function |
| $y$ | Query coordinate in the input domain of $s$ |
| $\mathcal{G} : \mathcal{X} \to \mathcal{Y}$ | The operator mapping between function spaces |
| $\mathcal{E} : \mathcal{X} \to \mathbb{R}^n$ | Encoder mapping |
| $\mathcal{D} : \mathbb{R}^n \to \mathcal{Y}$ | Decoder mapping |
| **Fourier Neural Operator** | |
| $\mathcal{F}, \mathcal{F}^{-1}$ | Fourier transform and its inverse |
| $\mathcal{F}_n, \mathcal{F}_n^{-1}$ | Discrete Fourier transform and its inverse truncated on the first $n$ modes |
| $K$ | Linear transformation applied to the $n$ leading Fourier modes |
| $W$ | Linear transformation (bias term) applied to the layer inputs |
| $k$ | Fourier modes / wave numbers |
| $\sigma$ | Activation function |
| **Continuous Vision Transformer** | |
| $\text{PE}_t$ | Temporal positional embedding |
| $\text{PE}_s$ | Spatial positional embedding |
| MSA | Multi-head self-attention |
| MHA | Multi-head attention |
| LN | Layer normalization |
| $P$ | Patch size of Vision Transformer |
| $C$ | Embedding dimension of Vision Transformer |
| $\mathbf{x} \in \mathbb{R}^{N_x \times N_y \times C}$ | Latent grid features |
| $\beta$ | Locality of interpolated latent grid features |
| **Partial Differential Equations** | |
| $\zeta$ | Vorticity of shallow water equations |
| $\eta$ | Height of shallow water equations |
| $\mathbf{u}$ | Velocity field |
| $P$ | Pressure field |
| $c$ | Passive scalar (smoke) |
| $\nu$ | Viscosity |
| **Hyperparameters** | |
| $B$ | Batch size |
| $Q$ | Number of query coordinates in each batch |
| $D$ | Number of latent variables of interest |
| $T$ | Number of previous time-steps |
| $H \times W$ | Resolution of spatial discretization |

## B    Neural Fields & Conditioned Neural Fields

A neural field is a function over a continuous domain parameterized by a neural network (Stanley, 2007). Recently, the computer vision community has adopted neural fields to represent 3D spatial fields, such as neural radiance fields (NeRFs) (Mildenhall et al., 2021), spatial occupancy fields (Mescheder et al., 2019), and signed distance functions (Park et al., 2019). These fields represent functions defined over continuous spatial domains, with NeRFs additionally taking an element of a continuous ray space as input. In this paper, a point in the domain of a neural field is referred to as a *query point*, here denoted by $y$. Conditioned neural fields allow these fields to change based on auxiliary inputs $u$, often represented as latent codes $z$, without the need to retrain the entire field; see Figure 7. The latent code affects the forward pass by modifying the parameters of the base field through either global conditioning, where the forward pass is affected uniformly across all query points, or local conditioning, where the modifications depend on the specific query point.

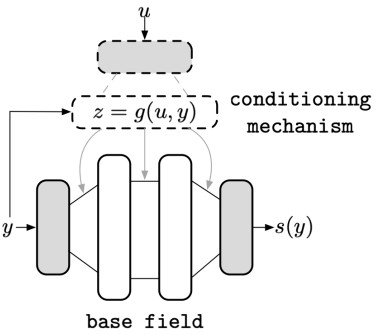

Figure 7: The structure of a *conditioned neural field* with both global and local conditioning.

**Global Conditioning.**    Global conditioning in a neural field refers to modifying the forward pass only by the input $u$ and in a uniform manner over the entire query domain. This can be achieved through concatenation, where a global latent code is appended to the query vector and passed together through an MLP (Chen & Zhang, 2019; Liu et al., 2021a; Jang & Agapito, 2021). Alternatively, a hypernetwork could accept the auxiliary input and determine the neural field's parameter values based on this input (Ha et al., 2017; Rebain et al., 2022). However, modifying every parameter of the base field for different auxiliary inputs can be computationally expensive. For example, feature-wise Linear Modulation (FiLM) (Perez et al., 2018), which originates from conditional batch normalization (De Vries et al., 2017), has been proposed to modify only the sine activation function parameters in a SIREN network (Sitzmann et al., 2020). This technique has been used for global conditioning in methods such as $\pi$-GAN (Chan et al., 2021).

**Local Conditioning.**    Local conditioning occurs when the latent code $z$, which modifies the base field's parameters, is a function of the input query $y$, i.e., $z = g(y)$. This results in query-dependent field parameters that can vary across the query domain. A common approach to generate local latent codes is to process the image or scene with a convolutional network and extract features from activations with receptive fields containing the query point. This can be done using 2D convolutions of projected 3D scenes (Xu et al., 2019; Yu et al., 2021; Trevithick & Yang, 2021) or 3D convolutions (Peng et al., 2020; Liu et al., 2020; Chibane et al., 2020). Muller *et al.* (Müller et al., 2022) learn features directly on multiresolution grid vertices and create local features via nearest grid point interpolation for each grid. Chan *et al.* (Chan et al., 2022) create features along three 2D planes and interpolate features from projections onto these planes. DeVries *et al.* (DeVries et al., 2021) learn latent codes over patches on a 2D "floorplan" of a scene. Other methods divide the query domain into patches or voxels and learn a latent code $z$ for each patch or voxel (Chabra et al., 2020; Tretschk et al., 2020; Chen et al., 2022).

## C    Neural Operators as Conditioned Neural Fields

Here we demonstrate that many popular operator learning architectures are in fact neural fields with local and/or global conditioning. While connections between architectures such as the DeepONet and the Fourier Neural Operator have been described previously (Kovachki et al., 2021), they are typically formulated under certain architecture choices which allow these architectures to approximate one another. In contrast, here we show that, by viewing these architectures as conditioned neural fields, they are explicitly related through their respective choices of conditioning mechanisms and base fields.

## C.1 ENCODER-DECODER NEURAL OPERATORS

A large class of operator learning architectures can be described as encoder-decoder architectures according to the following construction. To learn an operator $\mathcal{G} : \mathcal{X} \to \mathcal{Y}$, an input function $u \in \mathcal{X}$ are first mapped to a finite dimensional latent representation through the encoding map $\mathcal{E} : \mathcal{X} \to \mathbb{R}^n$. This latent representation is then mapped to a callable function through a decoding map $\mathcal{D} : \mathbb{R}^n \to \mathcal{Y}$. The composition of these two maps gives the architecture of the operator approximation scheme, $\mathcal{G} = \mathcal{D} \circ \mathcal{E}$. The class of encoder-decoder architectures includes the DeepONet (Lu et al., 2021), the PCA-based architecture of (Bhattacharya et al., 2021), and NoMaD (Seidman et al., 2022).

These encoder-decoder architectures can be seen as neural fields with a global conditioning mechanism by viewing the role of the decoder as an operator which performs the conditioning of a base field, with a latent code derived from a parameterized encoder, $z = \mathcal{E}_\phi(u)$.

**DeepONet.** For example, in a DeepONet, the base field is a neural network $t_\theta(y)$ (trunk network) appended with a last linear layer, and the decoder map conditions the weights of this last layer via the output of the encoding map $\mathcal{E}_\phi(u)$ (branch network); see Figure 8(a) for a visual illustration. In other words, the decoder map is a weighted linear combination of $n$ basis elements $t_\theta(y)$, where the weights are modulated by a global encoding of the input function $\mathcal{E}_\phi(u)$, yielding outputs of the form $s(y) = \langle \mathcal{E}_\phi(u), t_\theta(y) \rangle$. The PCA-based architecture of (Bhattacharya et al., 2021) can be viewed analogously using as basis elements the leading $n$ eigenfunctions of the empirical covariance operator computed over the output functions in the training set.

**NoMaD.** Seidman *et al.* (Seidman et al., 2022) demonstrated how encoder-decoder architectures can employ nonlinear decoders to learn low-dimensional manifolds in an ambient function space. These so-called NoMaD architectures typically perform their conditioning by concatenating the input function encoding with the query input, i.e., $s(y) = f_\theta(y, \mathcal{E}_\phi(u))$; see Figure 8(b) for an illustration.

## C.2 INTEGRAL KERNEL NEURAL OPERATORS

Integral Kernel Neural Operators form another general class of neural operators that learn mappings between function spaces by approximating the integral kernel of an operator using neural networks. They were introduced as a generalization of the Neural Operator framework proposed by Li *et al.* (Li et al., 2020). The key idea behind here is to represent the operator $\mathcal{G} : \mathcal{X} \to \mathcal{Y}$ as an integral kernel, $\mathcal{G}(u)(y) = \int_\Omega \kappa(x, y, u(x)) dx$, where $\kappa$ is a kernel function. In practice, the kernel function $\kappa$ is parameterized by a neural network, and the integral is then approximated using a quadrature rule, such as Monte Carlo integration, Gaussian quadrature, or via spectral convolution in the frequency domain.

**Graph Neural Operator (GNO).** A single layer of the GNO (Li et al., 2020) is given by

$$s(y) = \sigma\Big( Wu(y) + \frac{1}{|N(y)|} \sum_{x \in N(y)} \kappa_\phi(x, y, u(x)) u(x) \Big), \tag{10}$$

where $W \in \mathbb{R}^{d_u \times d_s}$ is a trainable weight matrix, and $\kappa_\phi(\cdot)$ is a local message passing kernel parameterized by $\phi$ (Li et al., 2020). As illustrated in Figure 8(c), the GNO layer is a conditioned neural field with local and global conditioning on the input function $u$. The base field uses a fixed positional encoding $\gamma(y)$ that maps query coordinates $y$ to input function values at neighboring nodes $x \in N(y)$, i.e., $\gamma(y) = u(x)/|N(y)|$, followed by a linear layer. The encoding width equals the maximum number of neighboring nodes, with zeros filling tailing entries for queries with fewer neighbors. The GNO layer is conditioned globally via the parameterized kernel $\kappa_\phi(\cdot)$ modulating the linear layer weights, and locally via a position-depended bias term $Wu(y)$ added as a skip connection. The layer outputs $s(y)$ are finally obtained by summing the linear layer outputs and the skip connection, then applying a nonlinear activation function $\sigma$.

**Fourier Neural Operator (FNO).** The FNO introduced a computationally efficient approach for performing global message passing in GNO layers assuming a stationary kernel $\kappa_\phi(\cdot)$ (Li et al., 2021; Bruna et al., 2014). Leveraging the Fourier transform, a single layer of FNO (Li et al., 2021) is given

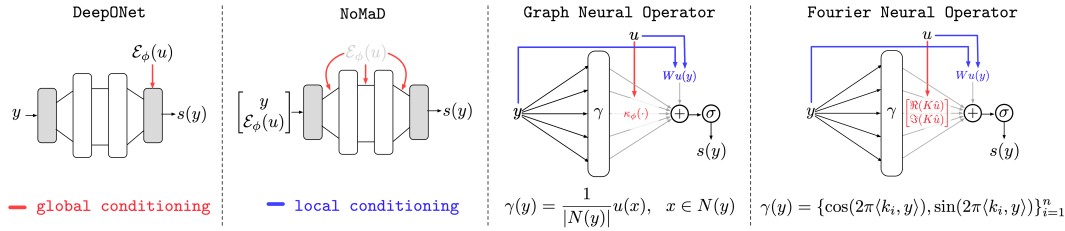

Figure 8: *Viewing common neural operators as conditioned neural fields.* Base field and conditioning mechanisms for: (a) DeepONet; (b) NoMaD; (c) the GNO layer; (d) the FNO layer.

by

$$s(y) = \sigma\Big(Wu(y) + \mathcal{F}_n^{-1}\left(K\mathcal{F}_n(u)\right)(y)\Big),\tag{11}$$

where $W \in \mathbb{R}^{d_o \times d_o}$ and $K \in \mathbb{R}^{n \times n}$ are trainable weight matrices, $\mathcal{F}_n : L^2 \to \mathbb{C}^n$ is the Fourier transform truncated to the first $n$ modes, and $\mathcal{F}_n^{-1} : \mathbb{C}^n \to L^2$ is the left inverse of this operator. We show that the FNO layer can be viewed as a conditioned neural field employing both local and global conditioning on the input function $u$. Without loss of generality, we demonstrate this for a one-dimensional query domain $y \in \mathbb{R}$. To this end, note that the convolutional part of the FNO layer (equation 11) can be re-written as (see Appendix D for a derivation)

$$\mathcal{F}_n^{-1}\left(K\mathcal{F}_n(u)\right)(y) = \sum_{j=1}^n \Re(K\hat{u})_j \cos(2\pi\langle k_j, y\rangle) - \sum_{j=1}^n \Im(K\hat{u})_j \sin(2\pi\langle k_j, y\rangle),\tag{12}$$

where $\hat{u} = \mathcal{F}_n u$ denotes the truncated Fourier transform of $u$, and $\Re(z)$ and $\Im(z)$ denote the real and imaginary parts of a complex vector $z \in \mathbb{C}^n$. We may interpret this transformation of $u(y)$ as a linear transformation (which depends on $\hat{u}$) acting on the positional encoding,

$$\gamma(y) = [\cos(2\pi\langle k_1, y\rangle), \sin(2\pi\langle k_1, y\rangle), \ldots, \cos(2\pi\langle k_n, y\rangle), \sin(2\pi\langle k_n, y\rangle)]^T.\tag{13}$$

The resulting expression (equation 12) is then acted on with a position dependant bias term $Wu(y)$, followed by a pointwise nonlinearity $\sigma$. In this manner, we see that an FNO layer is a neural field with a fixed positional encoding (first $n$ Fourier features), a global conditioning of the weights by the Fourier transform of $u$, and a local conditioning acting as a bias term by $Wu(y)$; see Figure 8(d) for an illustration. This neural field interpretation also suggests an alternative way for implementing FNO layers by explicitly using equation 12 to compute the inverse Fourier transform, thereby allowing them to be evaluated at arbitrary query points instead of a fixed regular grid.

**Extending to other neural operators.** In practice, GNO and FNO layers are stacked to form deeper architectures with a compositional structure. These can also be interpreted as conditioned neural fields, where the base field corresponds to the last GNO or FNO layer, while all previous layers are absorbed into the definition of the conditioning mechanism. Analogously, one can examine a broader collection of models that fall under the class of Integral Kernel Neural Operators, such as (Gupta et al., 2021; Tripura & Chakraborty, 2022; Cao et al., 2023; Fanaskov & Oseledets, 2023; Liu-Schiaffini et al., 2024).

## D  THE FNO LAYER AS A CONDITIONED NEURAL FIELD

**Proposition D.1.** *Suppose that a single layer of the Fourier Neural Operator (FNO) (Li et al., 2021) is given by*

$$s(y) = \sigma\Big(Wu(y) + \mathcal{F}_n^{-1}\left(K\mathcal{F}_n(u)\right)(y)\Big).\tag{14}$$

*Then the integral kernel term can be expressed as*

$$\mathcal{F}_n^{-1}\left(K\mathcal{F}_n(u)\right)(y) = \sum_{j=1}^n \Re(K\hat{u})_j \cos(2\pi\langle k_j, y\rangle) - \sum_{j=1}^n \Im(K\hat{u})_j \sin(2\pi\langle k_j, y\rangle),\tag{15}$$

*Proof.* Recall that the map $\mathcal{F}_n : L^2 \to \mathbb{C}^n$ is the Fourier transform truncated to the first $n$ modes and $\mathcal{F}^{-1} : \mathbb{C}^n \to L^2$ is the left inverse of this operator,

$$\mathcal{F}_n^{-1}(z)(y) = \sum_{j=1}^{n} z_j e^{2\pi i \langle k_j, y \rangle}. \tag{16}$$

Recall Euler's formula,

$$e^{2\pi i \langle k, y \rangle} = \cos(2\pi \langle k, y \rangle) + i \sin(2\pi \langle k, y \rangle). \tag{17}$$

When the (complex) Fourier coefficients $z_1, \dots, z_n$ come from a real valued signal, the imaginary components in the reconstruction equation 16 cancel and we are left with only real values. In this case, if we decompose each $z_j$ into real and imaginary parts, $z_j = \alpha_j + i\beta_j$, then using equation 17 we rewrite equation 16 as

$$\mathcal{F}_n^{-1}(z)(y) = \sum_{j=1}^{n} \alpha_j \cos(2\pi \langle k_j, y \rangle) - \sum_{j=1}^{n} \beta_j \sin(2\pi \langle k_j, y \rangle). \tag{18}$$

This allows us to view the convolutional part of the FNO layer equation 11 as

$$\mathcal{F}_n^{-1}\left(K\mathcal{F}_n(u)\right)(y) = \sum_{j=1}^{n} \Re(K\hat{u})_j \cos(2\pi \langle k_j, y \rangle) - \sum_{j=1}^{n} \Im(K\hat{u})_j \sin(2\pi \langle k_j, y \rangle), \tag{19}$$

where $\hat{u} = \mathcal{F}_n u$ denotes the truncated Fourier transform of $u$, and $\Re(z)$ and $\Im(z)$ denote the real and imaginary parts of a complex vector $z \in \mathbb{C}^n$.

$\square$

# E    THEORETICAL INSIGHTS ON COORDINATES EMBEDDINGS

Without loss of generality, we consider an MLP with scalar inputs and outputs. Specifically, let $x \in \mathbb{R}$ be the input coordinate, $g^{(0)}(x) = x$ and $d_0 = d_{L+1} = 1$. An MLP $f_\theta(x)$ is recursively defined as follows

$$\mathbf{f}_\theta^{(l)}(x) = \mathbf{W}^{(l)} \cdot \mathbf{g}^{(l-1)}(x) + \mathbf{b}^{(l)}, \quad \mathbf{g}^{(l)}(x) = \sigma(\mathbf{f}_\theta^{(l)}(x)), \quad l = 1, 2, \dots, L, \tag{20}$$

with a final linear layer defined by

$$f_\theta(x) = \mathbf{W}^{(L+1)} \cdot \mathbf{g}^{(L)}(x) + \mathbf{b}^{(L+1)}, \tag{21}$$

where $\mathbf{W}^{(l)} \in \mathbb{R}^{d_l \times d_{l-1}}$ is the weight matrix in $l$-th layer and $\sigma$ is an element-wise activation function. Here, $\theta = \left(\mathbf{W}^{(1)}, \mathbf{b}^{(1)}, \dots, \mathbf{W}^{(L+1)}, \mathbf{b}^{(L+1)}\right)$ represents all trainable parameters of the network. In particular, we suppose that the network is equipped with Tanh activation functions and all weight matrices are initialized by the Glorot initialization scheme (Glorot & Bengio, 2010), i.e., $\mathbf{W}^{(l)} \sim \mathcal{N}(0, \frac{2}{d_l + d_{l-1}})$ for $l = 1, 2, \dots, L + 1$. Moreover, all bias parameters are initialized to zeros.

**Definition E.1.** *A function $f : \mathbb{R}^n \to \mathbb{R}^m$ is called Lipschitz continuous if there exists a constant $L$ such that*

$$\|f(x) - f(y)\|_2 \le L\|x - y\|_2, \quad \forall x, y \in \mathbb{R}^n. \tag{22}$$

*The smallest $L$ for which the previous inequality is true is called the Lipschitz constant of $f$ and will be denoted $Lip(f)$.*

**Theorem E.2.** *((Federer, 2014), Theorem 3.1.6] If $f : \mathbb{R}^n \to \mathbb{R}^m$ is a locally Lipschitz continuous function, then $f$ is differentiable almost everywhere. Moreover, if $f$ is Lipschitz continuous, then*

$$Lip(f) = \sup_{x \in \mathbb{R}^n} \|\mathrm{D}_x f\|_2, \tag{23}$$

*where $\|M\|_2 = \sup_{\{x : \|x\|=1\}} \|Mx\|_2$ is the operator norm of the matrix $M \in \mathbb{R}^{m \times n}$.*

In particular, if $f$ is real valued (i.e. $m = 1$), its Lipschitz constant is the maximum norm of its gradient $Lip(f) = \sup_x \|\nabla f(x)\|_2$ on its domain set.

By Theorem 1 and the chain rule, the Lipschitz constant of MLPs have an explicit formula (Virmaux & Scaman, 2018; Latorre et al., 2020)

$$Lip(f_\theta) = \sup_{x \in \mathbb{R}^n} \left\| \prod_{i=L+1}^{2} \left( \mathbf{W}^{(i)} \cdot \mathrm{diag} \left( \dot{\sigma}\big(\mathbf{f}_\theta^{(i-1)}(x)\big) \right) \right) \cdot \mathbf{W}^{(1)} \right\|_2. \tag{24}$$

In Eq. equation 24, the diagonal matrices $\mathrm{diag}\left( \dot{\sigma}\big(\mathbf{f}_\theta^{(i-1)}(x)\big) \right)$ are difficult to evaluate, as they may depend on the input value $x$ and previous layers. Fortunately, most major activation functions are 1-Lipschitz. More specifically, these activation functions have a derivative $\dot{\sigma}(\cdot) \in [0, 1]$. Hence, we may replace the supremum on the input vector $x$ by a supremum over all possible values:

$$Lip(f_\theta) \leq \sup_{\mathbf{z}^{(i-1)} \in [0,1]^n} \left\| \prod_{i=L+1}^{2} \left( \mathbf{W}^{(i)} \cdot \mathrm{diag} \left( \mathbf{z}^{(i-1)} \right) \right) \cdot \mathbf{W}^{(1)} \right\|_2 \tag{25}$$

$$\leq \prod_{i=1}^{L+1} \left\| \mathbf{W}^{(i)} \right\|_2. \tag{26}$$

From the above derivation, we immediately obtain that the Lipschitz constant of a MLP is inherently tied to the weight matrices of its hidden layers. MLPs are known to suffer from spectral bias, struggling to learn high-frequency functions efficiently. One reasonable explanation is that $Lip(f)$ is bounded by the product of its parameter norms, which can only increase gradually during training by gradient descent. Consequently, higher frequency components of the target function are learned later in the optimization process (Rahaman et al., 2019). Random Fourier features (Tancik et al., 2020) have been proposed as a potential solution to this spectral bias issue. In the following theorem, we demonstrate that both random Fourier features and our proposed position embedding can substantially increase the network's Lipschitz constant at initialization. This enhancement enables more efficient learning of complex, fine-scale signals from the early stages of training.

**Theorem E.3.** *Given the definition of MLP in equation 20 and equation 21, the Lipschitz constant of the following coordinates embeddings can be bounded as follows*

(a) *For conventional MLPs, the coordinating embedding is trival and can be viewed as the first linear layer $\phi_1(x) = \mathbf{W}x + b$, where $\mathbf{W} \in \mathbb{R}^d$, $b \in \mathbb{R}$, and $d$ is the width of the network. Each element of $\mathbf{W}$ is independently sampled from $\mathcal{N}(0, \frac{1}{d})$. The expected Lipschitz constant of $\phi_1$ satisfies*

$$\mathbb{E}\,Lip(\phi_1) = \frac{\sqrt{2}}{\sqrt{d}} \frac{\Gamma((d+1)/2)}{\Gamma(d/2)} \approx 1 - \frac{1}{2d}. \tag{27}$$

(b) *The coordinate embedding of random Fourier features is given by $\phi_2(x) = \begin{bmatrix} \cos(\mathbf{W}x) \\ \sin(\mathbf{W}x) \end{bmatrix}$, where $\mathbf{W} \in \mathbb{R}^{d/2}$ and each element of $\mathbf{W}$ is independently sampled from $\mathcal{N}(0, s^2)$, with $s > 0$ being a hyperparameter. The expected Lipschitz constant of $\phi_2$ satisfies*

$$\mathbb{E}\,Lip(\phi_2) = 2s \cdot \frac{\Gamma\left(\frac{d/2+1}{2}\right)}{\Gamma(d/4)} \approx s\sqrt{d}\left(1 - \frac{1}{8d}\right). \tag{28}$$

(c) *The proposed coordinate embedding is defined as:*

$$\phi_\beta(x) = \sum_{i=1}^{n} \omega_i(x)\mathbf{f}_i = \sum_{i=1}^{n} e^{-\beta(x-x_i)^2}\mathbf{f}_i, \tag{29}$$

*where $\{x_i\}_{i=1}^{n}$ is a uniform grid in $[0, 1]$, and $\mathbf{f}_i \in \mathbb{R}^h$ are bounded latent features independent of coordinates, satisfying $\|\mathbf{f}_i\| \leq M$ for some $m, M > 0$. Let $h$ denote the spacing of the grid, i.e., $h = x_{i+1} - x_i$ for any $i \in 0, 1, ..., n-1$. For sufficiently large $\beta > 0$ such that $\frac{1}{\sqrt{2\beta}} < h$, the Lipschitz constant of $\phi_\beta$ is bounded as:*

$$M\sqrt{2\beta}e^{-\frac{1}{2}} \lesssim Lip(\phi_\beta) \lesssim 2M\sqrt{2\beta}e^{-\frac{1}{2}}. \tag{30}$$

*Proof.* (a) Let $\mathbf{W} = (W_i)_{i=1}^d$. Then we have

$$\text{Lip}(\phi_1) = \|\mathbf{W}\|_2 = \sqrt{\sum_{i=1}^d W_i^2} = \sqrt{\sum_{i=1}^d \frac{1}{d} Z_i^2} = \frac{1}{\sqrt{d}} \chi(d). \tag{31}$$

Since $\mathbb{E}\chi(d) = \sqrt{2}\frac{\Gamma((d+1)/2)}{\Gamma(d/2)}$,

$$\mathbb{E}\,\text{Lip}(\phi_1) = \frac{1}{\sqrt{d}}\mathbb{E}\chi(d) = \frac{\sqrt{2}}{\sqrt{d}}\frac{\Gamma((d+1)/2)}{\Gamma(d/2)}, \tag{32}$$

where $\Gamma(\cdot)$ is the Gamma function.

For large $d$, this can be approximated by:

$$\mathbb{E}\,\text{Lip}(\phi_1) \approx \frac{1}{\sqrt{d}}\sqrt{d}\left(1 - \frac{1}{4d}\right) = 1 - \frac{1}{4d}. \tag{33}$$

(b) We observe that

$$\text{Lip}(\phi_2) = \sup_{x \in R} \|\phi_2'\|_2 = \left\| \begin{bmatrix} \mathbf{W}\cos(\mathbf{Wx}) \\ -\mathbf{W}\sin(\mathbf{Wx}) \end{bmatrix} \right\|_2 \leq \sqrt{2\sum_{i=1}^{d/2} W_i^2} = \sqrt{2}\|\mathbf{W}\|_2. \tag{34}$$

Similar to the proof of (a), we have

$$\mathbb{E}\,\text{Lip}(\phi_2) \leq s\sqrt{2}\mathbb{E}\chi(d/2) = 2s \cdot \frac{\Gamma\left(\frac{\frac{d}{2}+1}{2}\right)}{\Gamma\left(\frac{d}{4}\right)}. \tag{35}$$

For large $d$, this can be approximated by:

$$\mathbb{E}\,\text{Lip}(\phi_2) \approx s\sqrt{2}\sqrt{\frac{d}{2}}\left(1 - \frac{1}{2d}\right) = s\sqrt{d}\left(1 - \frac{1}{2d}\right). \tag{36}$$

(c) To estimate the Lipschitz constant of $\phi$, we apply Theorem E.2, which reduces the problem to estimating its first-order derivative:

$$\phi_\beta'(x) = \sum_{i=1}^n \omega_i'(x)\mathbf{f}_i. \tag{37}$$

For any given $x$, there exists a unique $k$ such that $x \in [x_k, x_{k+1}]$. We can then decompose $\phi'(x)$ as follows:

$$\phi_\beta'(x) = \omega_k'(x)\mathbf{f}_k + \omega_{k+1}'(x)\mathbf{f}_{k+1} + \sum_{i \neq k, k+1} \omega_i'(x)\mathbf{f}_i := I_1 + I_2 + I_3. \tag{38}$$

Let us first estimate $I_1, I_2$. Since all $\mathbf{f}_i$ are bounded, we consider $g(x) = e^{-\beta x^2}$, as each $\omega_i(x)$ is a translation of $g$ by $x_i$. By differentiation, we obtain:

$$g'(x) = -2\beta x e^{-\beta x^2} \tag{39}$$

$$g''(x) = (4\beta^2 x^2 - 2\beta x)e^{-\beta x^2}. \tag{40}$$

Setting $g''(x) = 0$ yields $x = \pm\frac{1}{\sqrt{2\beta}}$. Consequently, $g'(x)$ achieves its maximum value of $\sqrt{2\beta}e^{-\frac{1}{2}}$ at $x = -\frac{1}{\sqrt{2\beta}}$ and its minimum value of $-\sqrt{2\beta}e^{-\frac{1}{2}}$ at $x = \frac{1}{\sqrt{2\beta}}$. Therefore, we can bound $\omega_i'(x)$ as follows:

$$-\sqrt{2\beta}e^{-\frac{1}{2}} \leq \omega_i'(x) \leq \sqrt{2\beta}e^{-\frac{1}{2}}. \tag{41}$$

To estimate $I_3$, recall that we assume $\frac{1}{\sqrt{2\beta}} < h$. For $i \neq k, k+1$ and fixed $x$, we have $mh \leq |x - x_i| \leq (m+1)h$ for a unique integer $m$, and $\omega_i'(x)$ decreases as $x$ moves away from $x_i$. Then:

$$|\omega_i'(x)| \leq 2\beta mh \cdot e^{-\beta m^2 h^2}. \tag{42}$$

Therefore, we have

$$\sum_{i \neq k, k+1} |\omega_i'(x)| \leq 2 \sum_{m=1}^{n} 2\beta m h \cdot e^{-\beta m^2 h^2} \tag{43}$$

$$\leq 2 \sum_{m=1}^{n} \int_{m-1}^{m} 2\beta h x \cdot e^{-\beta h^2 x^2} dx \tag{44}$$

$$= 2 \int_{0}^{n-1} 2\beta h x \cdot e^{-\beta h^2 x^2} dx \tag{45}$$

$$= \frac{2}{h} \left[ -e^{-\beta h x^2} \Big|_{0}^{n-1} \right] \tag{46}$$

$$= \frac{2 \left( 1 - e^{-\beta h^2 (n-1)^2} \right)}{h} \tag{47}$$

$$\leq \frac{2}{h}. \tag{48}$$

Since $\|\mathbf{f}_i\| \leq M$,

$$\left\| \phi_\beta'(x) \right\|_2 \leq M \left( |w_k'(x)| + |w_{k+1}'(x)| + \sum_{i \neq k, k+1} |\omega_i'(x)| \right) \tag{49}$$

$$\leq M \left( 2\sqrt{2\beta} e^{-\frac{1}{2}} + \frac{2}{h} \right). \tag{50}$$

Therefore, we conclude that

$$\mathrm{Lip}\left( \phi_\beta \right) \lesssim 2M\sqrt{2\beta} e^{-\frac{1}{2}}. \tag{51}$$

To obtain a lower bound for $\mathrm{Lip}(\phi_\beta)$, we proceed as follows: Without loss of generality, assume $\|\mathbf{f}_2\|_2 = M$. Let $x^* = h - \frac{1}{\sqrt{2\beta}}$, where $|\omega_2'|$ achieves its maximum. Then

$$\|\phi_\beta'(x^*)\|_2 \geq \|\omega_2'(x^*)\mathbf{f}_2\|_2 - \|\omega_1'(x^*)\mathbf{f}_1\|_2 - \sum_{i \geq 3} \|\omega_i'(x^*)\mathbf{f}_i\|_2 = J_1 - J_2 - J_3. \tag{52}$$

We can bound each term as follows:

$$J_1 = M\sqrt{2\beta} e^{-\frac{1}{2}}, \tag{53}$$

and

$$J_2 \leq M(2\beta) \left( h - \frac{1}{\sqrt{2\beta}} \right) e^{-\beta \left( h - \frac{1}{\sqrt{2\beta}} \right)^2}. \tag{54}$$

Note that $J_2$ goes to 0 as $\beta$ goes to infinity. So $J_2$ is bounded with respect to $\beta$. For $J_3$, we similarly estimate:

$$J_3 \leq M \sum_{m=1}^{n} \beta m h \cdot e^{-\beta m^2 h^2} \leq \frac{\left( 1 - e^{-\beta h^2 (n-1)^2} \right)}{h} \leq \frac{1}{h}. \tag{55}$$

Thus, combining these estimates, we conclude that

$$\mathrm{Lip}(\phi_\beta) \gtrsim M\sqrt{2\beta} e^{-\frac{1}{2}}. \tag{56}$$

$\square$

Our theorem yields key insights into the Lipschitz constants of various coordinate embedding methods. For trivial coordinate embeddings (linear layer MLP), the Lipschitz constant is approximately 1.

This has no impact on the network's global Lipschitz constant, resulting in the network remaining susceptible to spectral bias.

In the case of random Fourier features, the Lipschitz constant can be improved through two main approaches. First, by increasing the standard deviation of the Gaussian used for sampling the frequency matrix. Second, by increasing the network width. This observation implies a trade-off between network width and the frequencies used in the Fourier features.

For our proposed coordinate embeddings, the Lipschitz constant primarily depends on the parameter $\beta$ in the interpolation process. This dependence on a single parameter potentially offers more straightforward control over the Lipschitz constant compared to Fourier features.

These findings significantly impact neural field design and optimization using different coordinate embedding techniques. They provide a quantitative basis for comparing embedding methods' capabilities, particularly in addressing the spectral bias inherent in standard MLPs. The straightforward control of the Lipschitz constant in our proposed method (via $\beta$) may offer advantages in model tuning and optimization.

## F    EXPERIMENTAL DETAILS

### F.1    TRAINING AND EVALUATION

**Training recipe.**    We use a unified training recipe for all CViT experiments. We employ AdamW optimizer (Kingma & Ba, 2014; Loshchilov & Hutter, 2017) with a weight decay $10^{-5}$. Our learning rate schedule includes an initial linear warm-up phase of $5,000$ steps, starting from zero and gradually increasing to $10^{-3}$, followed by an exponential decay at a rate of $0.9$ for every $5,000$ steps. The loss function is a one-step mean squared error (MSE) between the model predictions and the corresponding targets at the next time-step, evaluated at randomly sampled query coordinates:

$$\text{MSE} = \frac{1}{B}\frac{1}{Q}\frac{1}{D}\sum_{i=1}^{B}\sum_{j=1}^{Q}\sum_{k=1}^{D}\left|\hat{s}_i^{(k)}(\mathbf{y_j}) - s_i^{(k)}(\mathbf{y_j})\right|_2^2, \tag{57}$$

where $s_i^{(k)}(\mathbf{y_j})$ denotes the $k$-th variable of the $i$-th sample in the training dataset, evaluated at a query coordinate $\mathbf{y}_j$, and $\hat{s}$ denotes the corresponding model prediction. It is worth noting that our objective function is a one-step loss, which differs significantly from the rollout loss used in training DPOTs All models are trained for $2 \times 10^5$ iterations with a batch size $B = 64$. Within each batch, we randomly sample $Q = 1,024$ query coordinates from the grid and corresponding output labels.

**Training instabilities**    We observe certain training instabilities in CViT models, where the loss function occasionally blows up, causing the model to collapse. To stabilize training, we clip all gradients at a maximum norm of 1. If a loss blowup is detected, we restart training from the last saved checkpoint.

**Evaluation.**    After training, we obtain the predicted trajectory by performing an auto-regressive rollout on the test dataset. We evaluate model accuracy using the relative $L^2$ norm, as commonly used in Li *et al.*(Li et al., 2021):

$$\text{Rel. } L^2 = \frac{1}{N_{\text{test}}}\frac{1}{D}\sum_{i=1}^{N_{\text{test}}}\sum_{k=1}^{D}\frac{\|\hat{s}_i^{(k)} - s_i^{(k)}\|_2}{\|s_i^{(k)}\|_2}, \tag{58}$$

where the norm is computed over the rollout prediction at all grid points, averaged over each variable of interest.

### F.2    MODEL DETAILS

**CViT.**    For the 1D advection equation, we create a tiny version of CViT to ensure the model size is comparable to other baselines. Specifically, we use a patch size of 4 to generate tokens. The encoder consists of 6 transformer layers with an embedding dimension of 256 and 16 attention heads. A single transformer decoder layer with 16 attention heads is used.

For the shallow water equation, compressible and incompressible Navier-Stokes equations and diffusion reaction equations, the CViT configurations are detailed in Table 1. While we use a patch size of $8 \times 8$ by default, we use a smaller patch size of $4 \times 4$ when training CViT-L for the Navier-Stokes problem. We find that using a smaller patch size better captures the small-scale features in flow motion, leading to improved accuracy.

**DeepONet.** For the DeepONet, the branch network (encoder) is an MLP with 4 hidden layers of 512 neurons each. The trunk-net (decoder) also employs an MLP with 4 hidden layers of 512 neurons.

**NoMaD.** The encoder consists of 4 hidden layers with 512 neurons each, mapping the input to a latent representation of dimension 512. The decoder, also an MLP with 4 hidden layers of 512 neurons, takes the latent representation and maps it to the output function.

**Dilated ResNet.** Following the comprehensive experiments and ablation studies in PDEArena (Gupta & Brandstetter, 2022), we adopt the optimal Dilated ResNet configuration reported in their work. This model has 128 channels and employs group normalization. It consists of four residual blocks, each comprising seven dilated CNN layers with dilation rates of [1, 2, 4, 8, 4, 2, 1].

**U-Nets.** Based on comprehensive experiments and ablation studies in PDEArena (Gupta & Brandstetter, 2022), we select the two strongest U-Net variants reported in their work: **U-Net$_{\text{att}}$** and **U-F2Net**. For both architectures, we use one embedding layer and one output layer with kernel sizes of $3 \times 3$. Besides, we use 64 channels and a channel multipliers of $(1, 2, 2, 4)$ and incorporate residual connections in each downsampling and upsampling block. Pre-normalization and pre-activations are applied, along with GELU activations. For U-Net$_{\text{att}}$, attention mechanisms are used in the middle blocks after downsampling. For U-F2Net, we replace the lower blocks in both the downsampling and upsampling paths of the U-Net architecture with Fourier blocks. Each Fourier block consists of 2 FNO layers with modes 16 and residual connections.

**UNO.** Following the comprehensive experiments and ablation studies in PDEArena (Gupta & Brandstetter, 2022), we adopt the optimal UNO of 128 channels.

**FNO.** For the 1D advection equation, our FNO model employs 3 Fourier neural blocks, each with 256 channels and 12 Fourier modes. The output is projected back to a 256-dimensional space before passing through a linear layer that produces the solution over the defined grid.

For the shallow water equation, we employ the best FNO model from comprehensive experiments and ablation studies in PDEArena (Gupta & Brandstetter, 2022). Specifically, we use the FNO consisting of 4 FNO layers, each with 128 channels and 32 modes. Additionally, we use two embedding layers and two output layers with kernel sizes of $1 \times 1$, as suggested in Li *et al.*(Li et al., 2021). We use GeLU activation functions and no normalization scheme.

For the Navier-Stokes equation, we follow the problem setup in DPOT (Hao et al., 2024), which differs from the setup in PDEArena (Gupta & Brandstetter, 2022). Therefore, we directly report the results from DPOT (Hao et al., 2024).

**FFNO / GK-T / OFormer / GNOT / DPOT / MPP.** We directly report the results from DPOT (Hao et al., 2024). Detailed implementations can be found in their paper.

### F.3 LINEAR ADVECTION EQUATION

We consider the one-dimensional linear advection equation

$$\frac{\partial u}{\partial t} + c \frac{\partial u}{\partial x} = 0, \qquad x \in [0, 1],$$

with periodic boundary conditions $u(0, t) = u(1, t)$ given an initial condition $u(x, 0) = u_0(x)$ with the advection velocity $c = 1$. To generate discontinuous initial conditions, the initial condition $u_0$ is assumed to be

$$u_0 = -1 + 2 \, \mathbb{1}_{\{\tilde{u_0} \geq 0\}},$$

Table 5: Performance of neural operator architectures under the relative L2 and Total Variation metrics for the linear advection with discontinuous initial conditions benchmark.

| Model | # Params | Rel. L2 error (%, ↓) | | | Total Variation error (↓) | | |
|---|---|---|---|---|---|---|---|
| | | Mean | Median | Worst-case | Mean | Median | Worst-case |
| FNO | 4.98M | 13.94 | 12.03 | 56.52 | 0.0406 | 0.0309 | 0.1910 |
| DeepONet | 2.20M | 13.77 | 12.20 | 39.64 | 0.0572 | 0.0453 | 0.2300 |
| NoMaD | 2.21M | 13.64 | 11.01 | 69.00 | **0.0326** | **0.0245** | 0.1732 |
| CViT | 3.03M | **11.70** | **10.68** | **35.44** | 0.0334 | 0.0274 | **0.1516** |

where $\tilde{u}_0$ a centered Gaussian,

$$\widetilde{u_0} \sim \mathcal{N}(0, C) \text{ and, } C = \left(-\Delta + \tau^2\right)^{-d}.$$

Here $-\Delta$ denotes the Laplacian on $D$ subject to periodic conditions on the space of spatial mean zero functions. Morevoer, $\tau$ denotes the inverse length scale of the random field and $d$ determines the regularity of $\widetilde{u_0}$. For this example, we set $\tau = 3$ and $d = 2$.

**Dataset.** We make use of the datasets released by Hoop *et al.*(de Hoop et al., 2022). This dataset consists of 40,000 samples of discretized initial conditions on a grid of $N = 200$ points.

**Problem setup.** We follow the problem setup by Hoop *et al.*(de Hoop et al., 2022). Our objective is to predict the solution profile at time $t = 0.5$. We compare the CViT model's performance against several strong baseline neural operators: DeepONet (Lu et al., 2021), NoMaD (Seidman et al., 2022) and Fourier Neural Operators (Li et al., 2021). For the training and evaluation, we considered a split of 20,000 samples used for training, 10,000 for validation, and 10,000 for testing.

**Training.** For all models, we use a batch size of 256 with a grid sampling size of 128, except for the FNO, which uses the full grid size of 200. The models are trained using the AdamW optimizer (Loshchilov & Hutter, 2017) with an exponential decay learning rate scheduler. The models are trained for 200,000 iterations, and the best model state is selected based on the validation set performance to avoid overfitting. The final performance is evaluated on the test set of 10,000 samples.

**Evaluation.** Due to the discontinuous nature of the problem, the relative L2 norm may not properly reflect the performance of the models. Instead, a more suitable metric for this task is the total variation (Stein & Shakarchi, 2009), defined as

$$TV(f, g) = \int_\Omega ||\nabla f| - |\nabla g||\mathrm{dx}.$$

The total variation quantifies the integrated difference between the magnitudes of the gradients of the predicted solution and the ground truth, providing a better indicator of a model's ability to capture and preserve discontinuities in the solution.

Table 5 presents a summary of our results. We notice that the CViT model achieves the lowest mean, median and worst-case relative $L^2$ errors, outperforming the other architectures on this metric. Furthermore, when considering the total variation metric, the CViT model exhibits competitive performance for the mean and median total variation values, and achieves the lowest prediction error the worst-case sample in the test dataset.

## F.4  SHALLOW-WATER EQUATIONS

Let $\mathbf{u}$ denote the velocity and $\eta$ interface height which conceptually represents pressure. We define $\zeta = \nabla \times \mathbf{u}$ and divergence $\mathcal{D} = \nabla \cdot \mathbf{u}$. The vorticity formulation is given by:

$$\frac{\partial \zeta}{\partial t} + \nabla \cdot (\mathbf{u}(\zeta + f)) = 0,$$

$$\frac{\partial \mathcal{D}}{\partial t} - \nabla \times (\mathbf{u}(\zeta + f)) = -\nabla^2 \left( \frac{1}{2} \left( u^2 + v^2 \right) + g\eta \right),$$

$$\frac{\partial \eta}{\partial t} + \nabla \cdot (\mathbf{u}h) = 0,$$

subject to the periodic boundary conditions. Here $f$ is Coriolis parameter and $g = 9.81$ is the gravitational acceleration, and $h$ is the dynamic layer thickness.

**Data generation.** The dataset is generated by PDEArena (Gupta & Brandstetter, 2022) using `SpeedyWeather.jl` on a regular grid with spatial resolution of $192 \times 96 \, (h = 1.875°, \Delta y = 3.75°)$, and temporal resolution of $\Delta t = 48 \, \text{h}$. The resulting dataset contains 6,600 trajectories with a temporal spatial resolution $11 \times 96 \times 192$.

**Problem setup.** We follow the problem setup reported in PDEArena (Gupta & Brandstetter, 2022). Our goal is to learn the operator that maps the vorticity and pressure fields from the previous 2 two time-steps to the next time-step. All models are trained with 5,600 trajectories and evaluated on the remain 1,000 trajectories. We report the relative $L^2$ errors over 5-steps rollout predictions.

**Training.** For training CViT models, please refer to section F.1. For training other baselines in Table 2, we follow the recommended training procedure and hyper-parameters presented in PDEArena (Gupta & Brandstetter, 2022) Specifically, we use the AdamW optimizer with a learning rate of $2 \times 10^{-4}$ and a weight decay of $10^{-5}$. Training is conducted for 50 epochs, minimizing the summed mean squared error. We use cosine annealing as the learning rate scheduler (Loshchilov & Hutter, 2016) with a linear warm-up phase of 5 epochs. An effective batch size of 32 is used for training.

**Impact of Grid Resolution.** In our experiments, we matched the resolution of the embedding grid to the dataset for convenience. However, our model's performance is not significantly affected as long as the embedding grid roughly matches the dataset resolution.

To validate this, we use CViT-B as our backbone architecture, conduct experiments with three different grid resolutions: $95 \times 191, 97 \times 193$ , and $128 \times 200$. We performed these experiments using the shallow water equation example under the same hyper-parameter settings. The resulting relative $L^2$ errors are summarized below:

Table 6: *Swallow Water equation:* Relative $L^2$ error for different grid resolutions.

| **Grid Resolution** | $96 \times 192$ | $95 \times 191$ | $97 \times 193$ | $128 \times 200$ |
|---|---|---|---|---|
| Rel. $L^2$ Error | $1.563 \times 10^{-2}$ | $1.572 \times 10^{-2}$ | $1.568 \times 10^{-2}$ | $1.562 \times 10^{-2}$ |

The results demonstrate negligible variation in error across different resolutions. This empirical evidence supports our claim that CViT's performance is robust to small variations in grid resolution, suggesting that precise matching between the embedding grid and dataset resolution is not critical for model efficacy.

**Impact of random Fourier feature frequencies.** We conduct an ablation study to investigate the impact of the scaling parameter $s$ in random Fourier feature embeddings. Using CViT-B as our backbone architecture, we trained models with different values of $s$ on the SWE benchmark. As shown in Table 7, model accuracy improves with moderately large values of $s$ but deteriorates when $s$ becomes too large (e.g., $s = 16\pi$), reverting to performance levels similar to $s = 4\pi$. Notably, our proposed grid-based embedding consistently outperforms the random Fourier feature approach across all tested values of $s$.

Figure 9: *Shallow water benchmark.* Ablation studies on the latent dimension of grid features and number of attention heads in decoder. Here we use CViT-B as our backbone architecture.

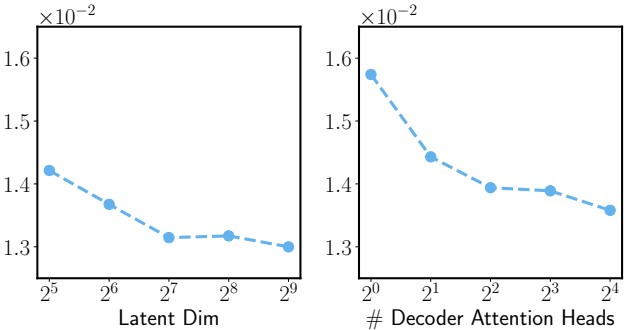

Table 7: *Shallow water benchmark.* Ablation studies on the hyper-parameter $s$ of random Fourier features. Here we use CViT-B as the backone.

|  | **Random Fourier Features** | | | | **Grid-based embedding** |
|---|---|---|---|---|---|
|  | $s = 2\pi$ | $s = 4\pi$ | $s = 8\pi$ | $s = 16\pi$ | $\beta = 10^5$ |
| Rel. $L^2$ error | $4.94 \times 10^{-2}$ | $4.16 \times 10^{-2}$ | $3.25 \times 10^{-2}$ | $4.08 \times 10^{-2}$ | $2.69 \times 10^{-2}$ |

### F.5 INCOMPRESSIBLE NAVIER-STOKES EQUATION

We consider the two-dimensional incompressible Navier-Stokes equations in the velocity-pressure formulation, coupled with a scalar field representing a transported particle concentration via the velocity field. The governing equations are given by:

$$\frac{\partial c}{\partial t} + \mathbf{u} \cdot \nabla c = 0,$$

$$\frac{\partial \mathbf{u}}{\partial t} + \mathbf{u} \cdot \nabla \mathbf{u} + \nabla p - \nu \nabla^2 \mathbf{u} = \mathbf{f},$$

$$\nabla \cdot \mathbf{u} = 0,$$

where $c$ is the scalar field, $\mathbf{u}$ is the velocity field, and and $\nu$ is the kinematic viscosity. In addition, the velocity field is affected through an external buoyancy force term $\mathbf{f}$ in the $y$ direction, represented as $\mathbf{f} = (0, f)^T$.

**Data generation.** The 2D Navier-Stokes data is generated by PDEArena (Gupta & Brandstetter, 2022), which is obtained on a grid with a spatial resolution of $128 \times 128 (h = 0.25, \Delta y = 0.25)$ and a temporal resolution of $\Delta t = 1.5$ s. The simulation uses a viscosity parameter of $\nu = 0.01$ and a

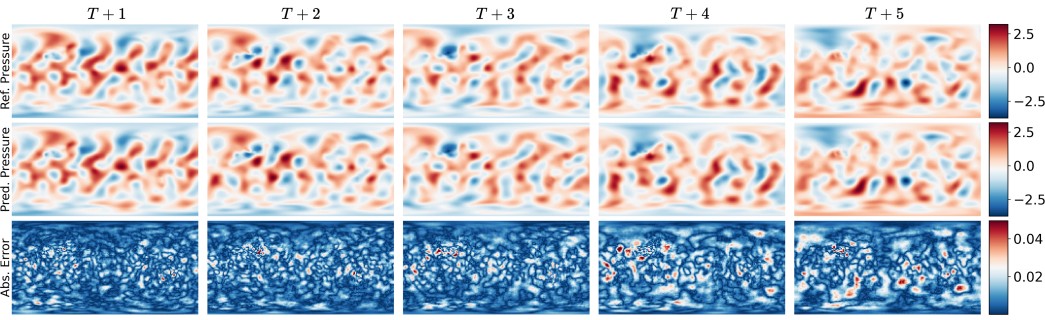

Figure 10: *Shallow water benchmark.* Representative CViT rollout prediction of the pressure field, and point-wise error against the ground truth.

buoyancy factor of $(0, 0.5)^T$. The equation is solved on a closed domain with Dirichlet boundary conditions $(v = 0)$ for the velocity, and Neumann boundaries $\left(\frac{\partial s}{\partial x} = 0\right)$ for the scalar field. The simulation runs for $21$ s, sampling every $1.5$ s . Each trajectory contains scalar and vector fields at 14 different time frames, resulting in a dataset with 7,800 trajectories.

**Problem setup.** We follow the problem setup reported in Hao *et al.* (Hao et al., 2024). We aim to learn the solution operator that maps the passive scalar and velocity fields from the previous 10 time-steps to the next time-step. The models are trained with 6,500 trajectories and tested on the remaining 1,300 trajectories. We report the resulting relative $L^2$ errors over 4-steps rollout predictions.

**Evaluation.** We directly report the results as presented in DPOT (Hao et al., 2024). For comprehensive details on the training procedures, hyper-parameters, please refer to their original paper.

**Impact of Latent Queries in the Time Aggregation Layer.** In our implementation, we use a single latent query in the Perceiver module. This query condenses information from $T$ time steps into a single time step representation, effectively reducing the length of spatial-temporal tokens and computational costs.

To investigate the effect of this hyperparameter on model performance, we conduct an ablation study by training CViT-L model with a patch size of 16, varying the number of latent queries $(n)$. As shown in Table 8, the results demonstrate that increasing the number of queries significantly enhances model performance, though it also increases the effective token length proportionally to the number of queries. This improvement likely stems from the fact that $n = 1$ discards too much useful information from the input trajectory during the time aggregation step, highlighting the potential for further optimization in this architecture.

Table 8: *Navier-Stokes benchmark.* Relative $L^2$ error for CViT-L models trained with a patch size of 16 and varying numbers of latent queries. Increasing the number of latent queries significantly improves accuracy. Note that these results differ from Table 3, where the best performance is achieved with a patch size of 4.

| # Latent queries | $n = 1$ | $n = 2$ | $n = 4$ | $n = 6$ |
|:---:|:---:|:---:|:---:|:---:|
| Rel. $L^2$ error | 6.56 % | 4.32 % | 3.81 % | 3.18 % |

**Analysis of cross-attention patterns.** we analyze the attention patterns that emerge in CViT when trained on the Navier-Stokes equation. We examine both the encoder and decoder cross-attention mechanisms to understand how the model learns to process temporal and spatial information.

The encoder's cross-attention mechanism is responsible for temporal aggregation. Figure 11 shows the averaged attention weights across different samples and attention heads, visualizing how the model attends to different timesteps in the input sequence. We observe that the attention weights systematically favor more recent timesteps, with the final timestep receiving notably higher attention. This temporal bias aligns with the physical intuition that recent states are more informative for predicting future dynamics while still maintaining some attention on earlier states for capturing longer-term dependencies.

The decoder's cross-attention patterns reveal how the model processes spatial relationships. Figure 12 visualizes the attention weights for various query points in the spatial domain. A clear pattern emerges where attention weights decay with distance from the query point, indicating that the model has learned to focus primarily on locally relevant information. This locality-aware behavior is particularly evident in regions of high fluid activity, suggesting the model adapts its attention patterns based on the underlying physics.

These attention patterns demonstrate that CViT learns physically meaningful representations through its cross-attention mechanisms. The encoder effectively prioritizes recent temporal information while the decoder captures spatially local dependencies, both of which are crucial for accurate fluid dynamics prediction.

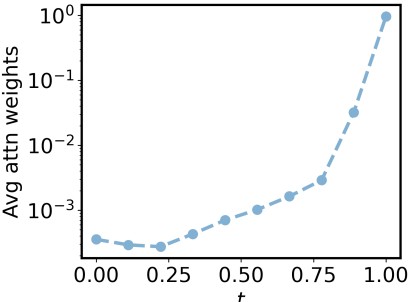

Figure 11: *Incompressible Navier-Stokes benchmark (NS).* Visualization of encoder cross-attention weights in CViT. It shows averaged attention weights across different samples and attention heads, where the x-axis represents input timesteps. The model learns to assign higher attention weights to more recent timesteps, particularly the final timestep.



Figure 12: *Incompressible Navier-Stokes benchmark (NS).* Visualization of decoder cross-attention patterns in CViT. Spatial attention weights for selected query points (marked by red stars), showing how attention naturally decays with distance from each query location. The attention patterns demonstrate that the model has learned to focus on spatially proximate regions when learning fluid dynamics.

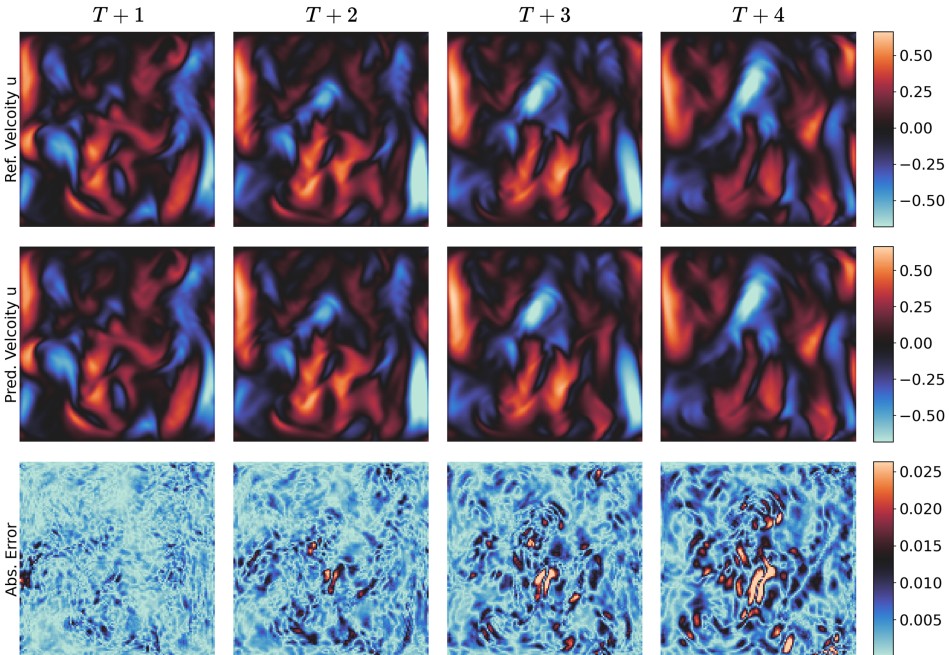

Figure 13: *Incompressible Navier-Stokes benchmark (NS).* Representative CViT rollout prediction of the velocity field in $x$ direction, and point-wise error against the ground truth.

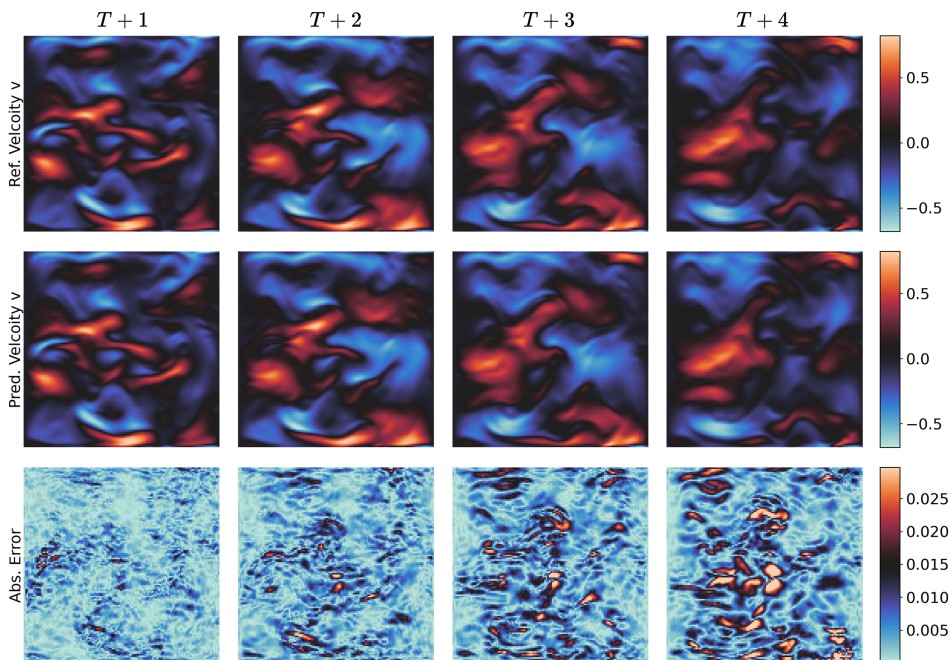

Figure 14: *Incompressible Navier-Stokes benchmark (NS).* Representative CViT rollout prediction of the velocity field in $y$ direction, and point-wise error against the ground truth.

### F.6 COMPRESSIBLE NAVIER-STOKES EQUATION

The 2D compressible Navier-Stokes equations govern fluid flow with variable density:

$$\partial_t \rho + \nabla \cdot (\rho \mathbf{v}) = 0, \tag{59}$$

$$\rho \left( \partial_t \mathbf{v} + \mathbf{v} \cdot \nabla \mathbf{v} \right) = -\nabla p + \eta \triangle \mathbf{v} + (\zeta + \eta/3)\nabla(\nabla \cdot \mathbf{v}), \tag{60}$$

$$\partial_t \left[ \epsilon + \frac{\rho v^2}{2} \right] + \nabla \cdot \left[ \left( \epsilon + p + \frac{\rho v^2}{2} \right) \mathbf{v} - \mathbf{v} \cdot \sigma' \right] = 0. \tag{61}$$

Here, $\rho$ denotes mass density, $\mathbf{v}$ velocity, $p$ gas pressure, $\Gamma = 5/3$, $\sigma'$ is the viscous stress tensor, and $\epsilon$ internal energy (determined by the equation of state), The viscous stress tensor $\sigma'$ incorporates both shear ($\eta$) and bulk ($\zeta$) viscosity coefficients. This system captures complex fluid dynamics phenomena, including shock wave dynamics, and finds applications in aerospace engineering and astrophysical fluid dynamics.

**Data generation.** We use the dataset generated by PDEBench (Takamoto et al., 2022), with $M = 0.1, \zeta = \eta = 0.01$, where $M = |v|/c_s$ is the Mach number, $c_s = \sqrt{\Gamma p/\rho}$ is the sound velocity.

**Problem setup.** We follow the problem setup reported in Hao *et al.* (Hao et al., 2024). We aim to learn the solution operator that maps the activator and inhibitor fields from the previous time-steps to the next time-step. The models are trained with 9,000 trajectories and tested on the remaining 1,000 trajectories. We report the resulting relative $L^2$ errors over rollout predictions.

**Training.** For training CViT models, please refer to section F.1. It is worth noting that for this specific task, we employ an initial learning rate of $5 \times 10^{-4}$, different from the $10^{-3}$ used in other experiments.

**Evaluation.** We directly report the results as presented in DPOT (Hao et al., 2024). For comprehensive details on the training procedures, hyper-parameters, please refer to their original paper.

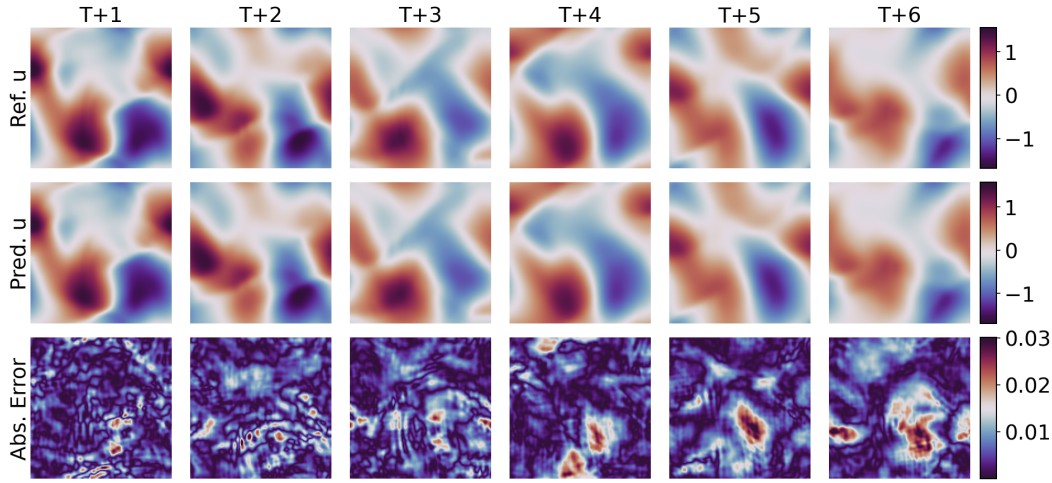

Figure 15: *Compressible Navier-Stokes Benchmark.* Representative CViT rollout prediction of the velocity field $u$, and point-wise error against the ground truth.

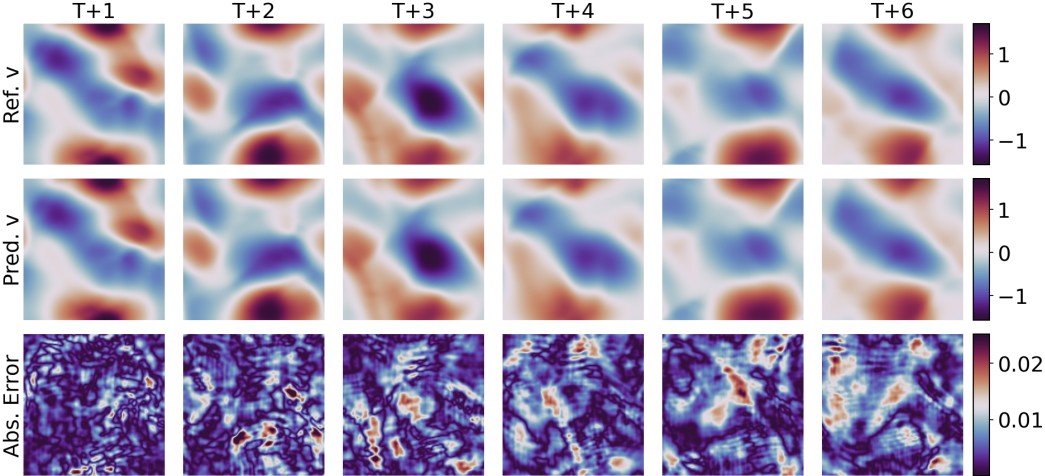

Figure 16: *Compressible Navier-Stokes Benchmark.* Representative CViT rollout prediction of the velocity field $v$, and point-wise error against the ground truth.

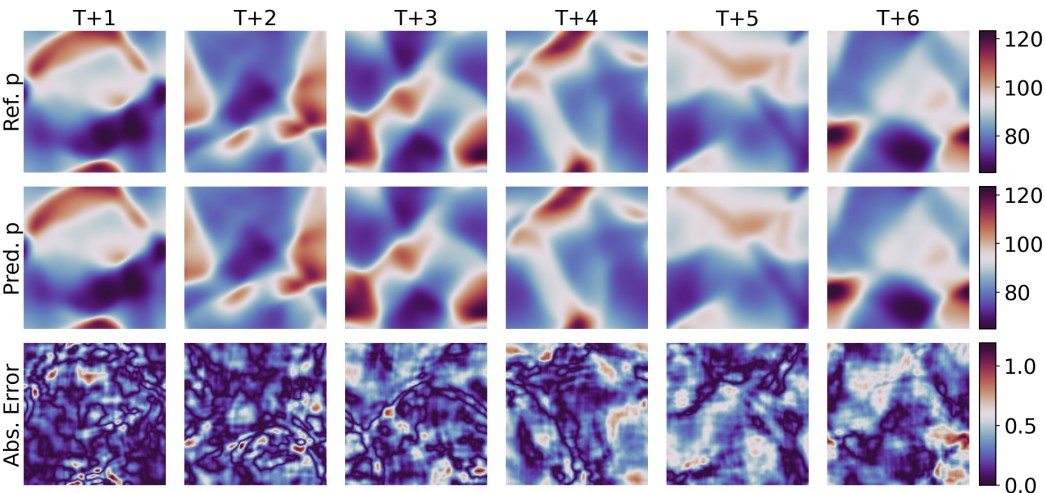

Figure 17: *Compressible Navier-Stokes Benchmark.* Representative CViT rollout prediction of the pressure field $p$, and point-wise error against the ground truth.

### F.7 DIFFUSION-REACTION EQUATION

The 2D diffusion-reaction equation is given by

$$\partial_t u = D_u \partial_{xx} u + D_u \partial_{yy} u + R_u,$$
$$\partial_t v = D_v \partial_{xx} v + D_v \partial_{yy} v + R_v,$$

where $u = u(t,x,y)$ is the the activator and $v = v(t,x,y)$ denotes the inhibitor. Here $D_u$ and $D_v$ are the diffusion coefficient for the activator and inhibitor, respectively, $R_u = R_u(u,v)$ and $R_v = R_v(u,v)$ are the activator and inhibitor reaction function, respectively. The domain of the simulation includes $x \in (-1,1), y \in (-1,1), t \in (0,5]$. The reaction functions for the activator and inhibitor are defined by:

$$R_u(u,v) = u - u^3 - k - v,$$
$$R_v(u,v) = u - v,$$

where $k = 5 \times 10^{-3}$, and the diffusion coefficients for the activator and inhibitor are $D_u = 1 \times 10^{-3}$ and $D_v = 5 \times 10^{-3}$, respectively. The initial condition is generated as standard normal random noise $u(0,x,y) \sim \mathcal{N}(0,1.0)$ for $x \in (-1,1)$ and $y \in (-1,1)$.

**Data generation.** We use the dataset generated by PDEBench (Takamoto et al., 2022), where discretized into $N_x = 512, N_y = 512$ and $N_t = 501$, as well as the downsampled version for the models training with $N_x = 128, N_y = 128$, and $N_t = 101$, the spatial discretization is performed using the finite volume method, and the time integration is performed using the built-in fourth order Runge-Kutta method in the scipy package.

**Problem setup.** We follow the problem setup reported in Hao *et al.* (Hao et al., 2024). We aim to learn the solution operator that maps the activator and inhibitor fields from the previous time-steps to the next time-step. The models are trained with 900 trajectories and tested on the remaining 100 trajectories. We report the resulting relative $L^2$ errors over rollout predictions.

**Training.** For training CViT models, please refer to section F.1. It is worth noting that for this specific task, we employ an initial learning rate of $5 \times 10^{-4}$, different from the $10^{-3}$ used in other experiments.

**Evaluation.** We directly report the results as presented in DPOT (Hao et al., 2024). For comprehensive details on the training procedures, hyper-parameters, please refer to their original paper.

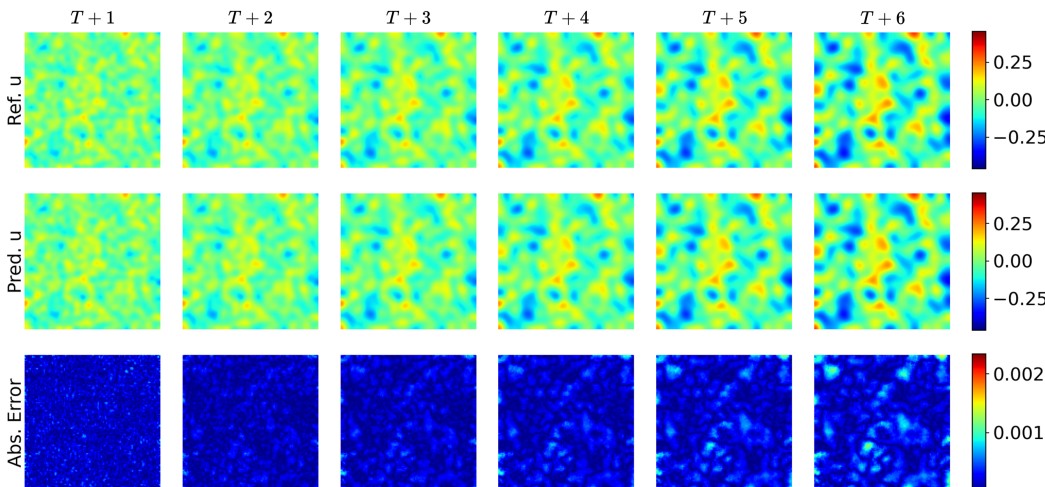

Figure 18: *Diffusion-reaction Benchmark.* Representative CViT rollout prediction of the density field $u$, and point-wise error against the ground truth.

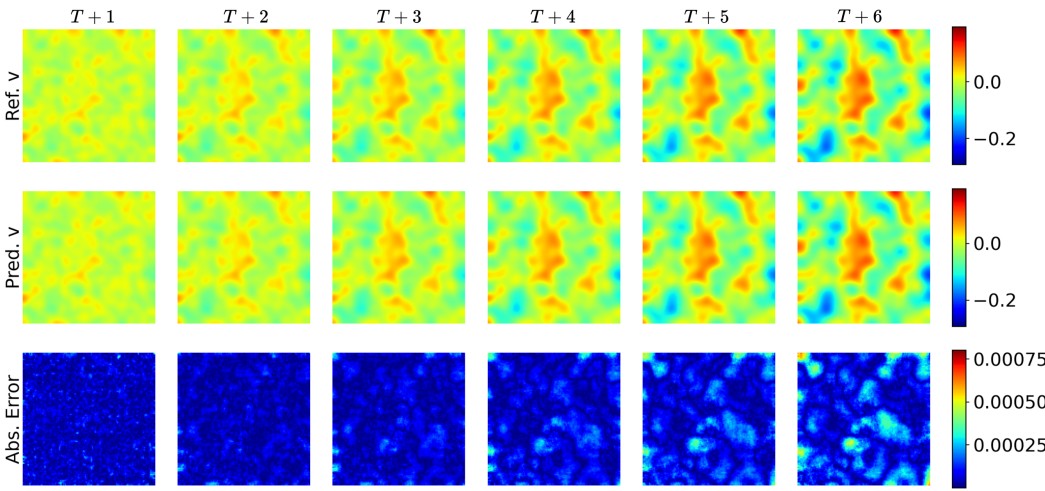

Figure 19: *Diffusion-reaction Benchmark.* Representative CViT rollout prediction of the density field $v$, and point-wise error against the ground truth.

## F.8 COMPUTATIONAL COST

All experiments were performed on a single Nvidia RTX A6000 GPU. Average training times varied between 5 hours and 60 hours, depending on the task, input resolution, model size, and patch size. The following table summarizes the training times for different models. We use a brief notation to indicate the model size and the input patch size: for instance, CViT-L/16 refers to the "Large" variant with a $16 \times 16$ input patch size.

Table 9: Training times (hours) on a single Nvidia RTX A6000 GPU for different models and patch sizes ($8 \times 8$, $16 \times 16$, $32 \times 32$) on benchmarks of Shallow water equation and Navier-Stokes equation.

| Method | Shallow Water | Navier-Stokes |
|---|---|---|
| CViT-S/8 | 15 | 9 |
| CViT-B/8 | 25 | 14 |
| CViT-L/8 | 57 | 28 |
| CViT-L/16 | 17 | 10 |
| CViT-L/32 | 9 | 6 |

Table 10: Comparison of parameter count and inference time. All measurements are performed on a single NVIDIA A6000 GPU with batch size 32, averaged over 100 runs.

| Model | # Params | Inference time (ms) |
|---|---|---|
| CViT-L | 93 M | 79.09 |
| ViT | 87 M | 45.49 |
| FNO | 67 M | 34.96 |
| UNet | 124 M | 44.25 |

