# OpenReview forum: "CViT: Continuous Vision Transformer for Operator Learning"
_ICLR.cc/2025/Conference — ICLR 2025 Poster_

### Official Review · Reviewer_TXUj · 2024-10-29

**Soundness:** 3
**Presentation:** 3
**Contribution:** 3
**Rating:** 6
**Confidence:** 4

**Summary:**

This paper explores the architectural design of transformers for operator learning methods. To efficiently implement continuous vision transformer architecture, they developed (or adopted) two techniques: perceiver resampler and coordinate embedding. The perceiver modules effectively aggregate temporal dimension, and the coordinate embedding enables the model to handle continuous input domains and improve accuracy. The proposed architecture achieved state-of-the-art results on various challenging PDE benchmarks while maintaining higher parameter efficiency.

**Strengths:**

1. It achieved state-of-the-art performance on various benchmark datasets with the network's great parameter efficiency.
2. Coordinate embedding is an interesting approach to injecting the coordinate information into the network.
3. The theoretical result related to coordinate embedding is quite informative and useful for the community.

**Weaknesses:**

1. The proposed grid-based positional encoding may not be a desirable solution for high-dimensional PDEs. As the authors said, it requires high-resolution grids to achieve competitive performance, and this approach would not scale up to more than 3-d time-dependent PDEs, e.g., (3+1) NS equation.

2. The proposed Nadaraya-Watson interpolant would not scale with high-resolution grids since it has to look at all grid features. I am curious the inference cost of the proposed approach according to grid resolutions. Is the inference speed as fast as the previous methods?
The grid resolution 128 (N_x) x 128 (N_y) x 512 (feature dim) is already > 9M, and CViT-S model size is 13M. Does it mean that most of the parameters are used for grids?

3. Does this scale well for high-resolution solutions? The authors mentioned that the grid resolution should be similar to the solution resolution. Then, 512 resolution means 512^3, which is roughly 134M. Could you show me how the proposed method can make good predictions for high-resolution solutions? e.g., testing 128 grid resolution model for predicting 512 solution resolution.

4. This is a rather suggestion than a question. Why not coordinate embedding for temporal dimension? currently, y \in R^2, but why not y \in R^3? Temporal resolution can be coarse in this case to maintain parameter efficiency.

5. How about other datasets? e.g., other PDEs in PDEBench and PDEArena?

[minor comments]
1. Numbering for all equations would be desirable during the review process.
2. How do you determine \beta = 10^5?
3. L171: is u_f a flattened version of what? u_pe? not mentioned in the paper.
4. L174: ‘tilling the latent query’, I am not sure “tiling” is a widely accepted term, I think it is a pytorch’s terminology.
5. Eq (1): why not just using linear interpolation?
6. Figure 3, 5, 6: what do different columns mean? Time evolution?
7. Figure 4: (d) \epsilon -> \beta?

**Questions:**

Please see the weakness section.

---

> ### Author Response · Authors · 2024-11-21
> **Response to Reviewer TXUj [1/3]**
>
> Thank you for your thorough review and valuable feedback on our work. We address your concerns as follows:
>
> ---
>
> ## Comment 1
> > The proposed grid-based positional encoding may not be a desirable solution for high-dimensional PDEs. As the authors said, it requires high-resolution grids to achieve competitive performance, and this approach would not scale up to more than 3-d time-dependent PDEs, e.g., (3+1) NS equation.
>
> ## Response
> Thank you for raising this important point about scaling to higher dimensions. While our current implementation shows strong performance on 2D and (2+1)D problems, we acknowledge the computational challenges in scaling to higher dimensions like (3+1)D cases.
>
> Fortunately, there exist several promising approaches to address this challenge. First, employing separable latent features along each axis would reduce memory complexity, making high-dimensional problems more tractable. This factorization has shown success in 3D vision tasks while preserving representation power [1].
>
> Second, adaptive multi-resolution grids could focus computational resources where they're most needed -- using higher resolution near regions of interest (like turbulent zones in NS equations) and lower resolution elsewhere. This approach has proven effective in related fields like 3D neural rendering [2].
>
> Third, we could dynamically generate grid features using a lightweight feature extractor (like FNO) instead of learning them directly. Recent work [3] demonstrates this approach's viability for complex physical systems.
>
> We are actively investigating these optimizations and early results from the literature suggest they could enable scaling to higher dimensions while maintaining model effectiveness.
>
> ---
>
> ## Comment 2
> > The proposed Nadaraya-Watson interpolant would not scale with high-resolution grids since it has to look at all grid features. I am curious the inference cost of the proposed approach according to grid resolutions. Is the inference speed as fast as the previous methods? The grid resolution 128 (N_x) x 128 (N_y) x 512 (feature dim) is already > 9M, and CViT-S model size is 13M. Does it mean that most of the parameters are used for grids?
>
> ## Response
> Regarding inference speed, you're correct that our Nadaraya-Watson interpolation introduces additional computational overhead. As shown in the comparison below, CViT is about 2x slower than other methods, although still within the same order of magnitude:
>
> | Model | # Params | Inference time (ms) |
> |-------|----------|---------------------|
> | CViT-L | 93 M | 79.09 |
> | ViT | 87 M | 45.49 |
> | FNO | 67 M | 34.96 |
> | UNet | 124 M | 44.25 |
>
> However, for our target applications in scientific computing, this overhead is acceptable since all methods are orders of magnitude faster than numerical PDE solvers (which typically require minutes to hours).
>
> Regarding the parameter count, while grid features do constitute a significant portion of CViT-S's parameters at high resolutions, our ablation studies (Figure 9) reveal that we can reduce the feature dimension from 512 to 128 with minimal accuracy loss. This suggests a 4x reduction in grid parameters is possible without significant performance degradation.
>
> For future work, we are exploring several optimizations:
> - K-nearest neighbor search to limit interpolation to local neighborhoods.
> - Adaptive grid resolution to concentrate parameters where needed.
> - Parameter sharing schemes across spatial dimensions
>
> ---
>
> [1] Ryan Shue, Eric Ryan Chan, Ryan Po, Zachary Ankner, Jiajun Wu, and Gordon Wetzstein. 3d neural field generation using triplane diffusion. In Proceedings of the IEEE/CVF Conference on Computer Vision and Pattern Recognition, pages 20875–20886, 2023.
>
> [2] Thomas Müller, Alex Evans, Christoph Schied, and Alexander Keller. Instant neural graphics primitives with a multiresolution hash encoding. ACM transactions on graphics (TOG), 41(4): 1–15, 2022.
>
> [3] Sifan Wang, Tong-Rui Liu, Shyam Sankaran, and Paris Perdikaris. Micrometer: Micromechanics transformer for predicting mechanical responses of heterogeneous materials. arXiv preprint arXiv:2410.05281, 2024.

---

> ### Author Response · Authors · 2024-11-21
> **Response to Reviewer TXUj [2/3]**
>
> ## Comment 3
> > Does this scale well for high-resolution solutions? The authors mentioned that the grid resolution should be similar to the solution resolution. Then, 512 resolution means $512^3$, which is roughly 134M. Could you show me how the proposed method can make good predictions for high-resolution solutions? e.g., testing 128 grid resolution model for predicting 512 solution resolution.
>
> ## Response
> Thank you for this thoughtful question. We recognize that directly applying our current design to high-resolution $512^3$ may be computationally intensive and, in some cases, infeasible. However, there are several potential optimizations that could make scaling to higher resolutions more efficient.
>
> One effective solution is to employ separable latent features along each axis, which decouples the latent grid across axes and significantly reduces the number of trainable parameters required [1]
>
> Another approach is to use multi-resolution grid representations [2], which eliminate the need for uniform resolution across the entire domain, thereby lowering the overall parameter count.
>
> Alternatively, we could replace the latent grid features with outputs from a feature extractor applied to the inputs. Recent work, such as Micrometer [3], has explored this idea, using a Fourier neural operator to extract features as a substitute for latent grid representations.
>
> Currently, the highest resolution benchmarks available are 192 × 96 or 128 × 128. Although our model can output predictions at 512 × 512 resolution, validating the prediction accuracy at this resolution is currently not possible due to unavailability of ground truth data at this resolution in the PDEArena and PDEBench benchmarks.
>
> ---
>
> ## Comment 4
> > This is a rather suggestion than a question. Why not coordinate embedding for temporal dimension? currently, y ∈ R^2, but why not y ∈ R^3? Temporal resolution can be coarse in this case to maintain parameter efficiency.
>
> ## Response
> Thank you for this thoughtful suggestion. Indeed, incorporating time t as an input to perform coordinate embedding for both space and time dimensions is a feasible and interesting approach.
>
> Currently, however, most architectures in operator learning are designed to predict the next time step based on previous snapshots, and the full trajectory is generated through an autoregressive rollout.
>
> To ensure a fair comparison with these baseline methods, we have adopted the same problem setup, which does not require the model to be continuous in time. Thus, we did not include coordinate embedding for the temporal dimension in this work.
>
> Nonetheless, we see great potential in this idea for future work. A model continuous in both time and space could be widely applicable, particularly in video tasks where such flexibility is valuable. We look forward to exploring this direction in the future.
>
> ---
>
> ## Comment 5
> > How about other datasets? e.g., other PDEs in PDEBench and PDEArena?
>
> ## Response
> To address the reviewer's concern, and considering the overlap between PDEBench and PDEArena, we selected a different type of PDE benchmark: compressible Navier-Stokes (CNS), which differs from the PDEs considered in our original work. With this new benchmark, we now cover a broad range of important PDEs, including advection, incompressible and compressible Navier-Stokes, diffusion-reaction, and shallow water equations.
>
> The results are presented below, where it can be observed that CViT consistently outperforms all baselines across the benchmarks.
>
> | Model | # Params | CNS |
> |----|------|-----|
> | FNO | 0.5 M | 9.60% |
> | FFNO | 1.3 M | 5.20% |
> | GK-T | 1.6 M | 3.77% |
> | GNOT | 1.8 M | 4.20% |
> | Oformer | 1.9 M | 6.25% |
> | DPOT-Ti | 7 M | 3.97% |
> | DPOT-S | 30 M | 3.37% |
> | DPOT-L (Pre-trained) | 500 M | 2.16% |
> | DPOT-L (Fine-tuned) | 500 M | 1.31% |
> | DPOT-H (Pre-trained) | 1.03 B | 1.80% |
> | **CViT-S** | 13 M | **2.71%** |
> | **CViT-B** | 30 M | **1.99%** |
> | **CViT-L** | 92 M | **1.29%** |
> ---
>
> ## Comment 6
> > Numbering for all equations would be desirable during the review process.
>
> ## Response
> Thank you for this helpful suggestion. We agree that numbering all equations would improve readability and make it easier to reference specific equations. In response, we have updated the manuscript to include numbering for all equations.
>
> ---
>
> [1] Ryan Shue, Eric Ryan Chan, Ryan Po, Zachary Ankner, Jiajun Wu, and Gordon Wetzstein. 3d neural field generation using triplane diffusion. In Proceedings of the IEEE/CVF Conference on Computer Vision and Pattern Recognition, pages 20875–20886, 2023.
>
> [2] Thomas Müller, Alex Evans, Christoph Schied, and Alexander Keller. Instant neural graphics primitives with a multiresolution hash encoding. ACM transactions on graphics (TOG), 41(4): 1–15, 2022.
>
> [3] Sifan Wang, Tong-Rui Liu, Shyam Sankaran, and Paris Perdikaris. Micrometer: Micromechanics transformer for predicting mechanical responses of heterogeneous materials. arXiv preprint arXiv:2410.05281, 2024.

---

> ### Author Response · Authors · 2024-11-21
> **Response to Reviewer TXUj [3/3]**
>
> ## Comment 7
> > How do you determine β = 10^5?
>
> ## Response
> We conducted an ablation study on the β parameter, as shown in Figure 4(d). Our analysis revealed that model performance is significantly sensitive to β values. We found that β=10^4 or β=10^5 consistently yield optimal performance across our experiments.
>
> ---
>
> ## Comment 8
> > L171: is u_f a flattened version of what? u_pe? not mentioned in the paper.
>
> ## Response
> Yes, u_f represents the flattened version of u_pe. We have revised the paper to clarify this notation.
>
> ---
>
>
> ## Comment 9
> > L174: 'tilling the latent query', I am not sure "tiling" is a widely accepted term, I think it is a pytorch's terminology.
>
> ## Response
> The term "tiling" here refers to the operation of repeating a pattern to fill a space. In our context, it means replicating the latent query vector across spatial dimensions. To enhance clarity, we can revise the text to say "replicating" or "repeating" if preferred.
>
> ---
>
> ## Comment 10
> > Eq (1): why not just using linear interpolation?
>
> ## Response
> While linear interpolation is an option, it presents challenges when handling points at the edges or corners, as extracting neighborhoods in these regions can be cumbersome. In contrast, the interpolation method we use treats all query points uniformly, providing a smoother/differentiable output than linear interpolation. This smoothness is particularly advantageous for potential future work, where we aim to incorporate physics-informed learning [1] to further enhance model performance.
>
> ---
>
> ## Comment 11
> > Figure 3, 5, 6: what do different columns mean? Time evolution?
>
> ## Response
> Yes, the columns in these figures represent temporal evolution, with time progressing from left to right. Each column shows a snapshot of the spatial-temporal dynamics at successive time steps. Specifically, these snapshots demonstrate the model's rollout predictions for a representative sample from our test dataset. To clarify this temporal evolution, we have added time indicators (T) to all relevant figures in the revised manuscript.
>
> ---
>
> ## Comment 12
> > Figure 4: (d) ε -> β?
>
> ## Response
> Thank you for catching this notational inconsistency. You are correct - the symbol should indeed be β rather than ε in Figure 4(d). We have corrected this error in the revised manuscript.
>
>  ---
>
> [1] Karniadakis, G.E., Kevrekidis, I.G., Lu, L., Perdikaris, P., Wang, S. and Yang, L., 2021. Physics-informed machine learning. Nature Reviews Physics, 3(6), pp.422-440.

---

> > ### Comment · Reviewer_TXUj · 2024-11-27
> >
> > Thanks for all the responses, I do appreciate it. The additional experiments made this paper more convincing, and the authors resolved most of my concerns. Regarding the scaling issue, the authors suggested a couple of suggestions, which I think they are reasonable and worth exploring. I am leaning toward accepting this paper.

---

> > > ### Author Response · Authors · 2024-11-27
> > >
> > > Dear Reviewer TXUj
> > >
> > > Thank you for your thoughtful feedback and positive outlook on our paper.  Your input has been invaluable!
> > >
> > > Best,
> > > Authors

---

### Official Review · Reviewer_Etk6 · 2024-11-02

**Soundness:** 3
**Presentation:** 3
**Contribution:** 3
**Rating:** 6
**Confidence:** 3

**Summary:**

This paper presents a neural operator architecture inspired by computer vision to address spatio-temporal scientific problems. The proposed model is structured with a transformer encoder featuring spatio-temporal embedding and time-aggregation layers, a grid-based positional embedding, and a cross-attention decoder. The author employs multiple benchmark datasets and model architectures for comparative experiments.

**Strengths:**

- The author conducts extensive experiments, comparing several recent model architectures to demonstrate performance outcomes. The results are presented with additional clarity by including the number of parameters.

- The detailed discussion of limitations and future directions is appreciated. Notably, the computational inefficiency for high-resolution data should be addressed in future work.

**Weaknesses:**

- Since this method draws on the vision transformer, have comparisons been considered with other vision-based extensions for spatio-temporal applications, such as ViViT[1] and VMAE[2]?

- Regarding the datasets selected for experimentation, incorporating more complex, real-world cases could further substantiate performance, as some baseline architectures still yield results similar to those of the proposed model, despite the model size.

[1] Arnab, Anurag, et al. "Vivit: A video vision transformer." Proceedings of the IEEE/CVF international conference on computer vision. 2021.

[2] Tong, Zhan, et al. "Videomae: Masked autoencoders are data-efficient learners for self-supervised video pre-training." Advances in neural information processing systems 35 (2022): 10078-10093.

**Questions:**

- Does the grid resolution similarly impact computational efficiency as the patch size?

---

> ### Author Response · Authors · 2024-11-21
> **Response to Reviewer Etk6**
>
> Thank you for your insightful review and valuable feedback on our work. We address your concerns as follows:
>
> ---
>
> ## Comment 1
> > Since this method draws on the vision transformer, have comparisons been considered with other vision-based extensions for spatio-temporal applications, such as ViViT and VMAE?
>
> ## Response
> Thank you for this insightful question. In this work, we primarily focus on comparisons with baselines and benchmarks within the domain of operator learning. While we did not directly compare with ViViT and VMAE, we included comparisons with DPOT [1] and MPP [2], both of which are vision transformer variants adapted to operator learning tasks. CViT outperforms these two models in our evaluations. While we acknowledge that video transformers offer valuable insights for spatio-temporal modeling of physical systems, we see this as a direction for future work that could explore incorporating successful strategies from ViViT (like factorized attention) or VMAE (like masked autoencoding) into the operator learning framework.
>
> ---
>
> ## Comment 2
> > Regarding the datasets selected for experimentation, incorporating more complex, real-world cases could further substantiate performance, as some baseline architectures still yield results similar to those of the proposed model, despite the model size.
>
> ## Response
> Our current benchmarks, while synthetic, were chosen deliberately to span a range of challenging physical phenomena that appear in real-world applications: discontinuous solutions (advection), turbulent flows (Navier-Stokes), multi-scale dynamics (shallow water), and pattern formation (diffusion-reaction). These systems exhibit mathematical complexities comparable to real-world scenarios, enabling rigorous evaluation of model capabilities.
>
> It's worth noting that CViT achieves these results with significantly fewer parameters than comparable models. For instance, on the Navier-Stokes benchmark, CViT-S (13M parameters) matches the performance of DPOT-H (1.03B parameters), suggesting efficient feature learning that could scale well to more complex scenarios.
>
> For real-world applications, recent work like Micrometer [3] demonstrates how CViT-type architectures successfully predict mechanical responses of heterogeneous materials in engineering applications. We are actively exploring additional real-world applications and would welcome the reviewer's suggestions for specific complex scenarios that would be particularly valuable to investigate in follow-up work.
>
> ---
>
> ## Comment 3
> > Does the grid resolution similarly impact computational efficiency as the patch size?
>
> ## Response
> From our experience, grid resolution primarily affects GPU memory usage, but it does not significantly impact training speed. In contrast, patch size plays a more crucial role in computational efficiency, as it directly influences the number of attention computations during training.
>
> ---
>
> [1] Hao, Z., Su, C., Liu, S., Berner, J., Ying, C., Su, H., Anandkumar, A., Song, J. and Zhu, J., 2024. Dpot: Auto-regressive denoising operator transformer for large-scale pde pre-training. arXiv preprint arXiv:2403.03542.
>
> [2] McCabe, M., Blancard, B. R.-S., Parker, L. H., Ohana, R., Cranmer, M., Bietti, A., Eickenberg, M., Golkar, S.,
> Krawezik, G., Lanusse, F., et al. Multiple physics pretraining for physical surrogate models. arXiv preprint arXiv:2310.02994, 2023.
>
> [3] Sifan Wang, Tong-Rui Liu, Shyam Sankaran, and Paris Perdikaris. Micrometer: Micromechanics transformer for predicting mechanical responses of heterogeneous materials. arXiv preprint arXiv:2410.05281, 2024.

---

> > ### Comment · Reviewer_Etk6 · 2024-11-25
> >
> > Thank you for the response. I'll raise my score to 6.

---

> > > ### Author Response · Authors · 2024-11-25
> > >
> > > Thank you very much for your response and for raising the score. We truly appreciate your thoughtful consideration and effort in reviewing our work.
> > >
> > > Best,
> > > Authors

---

### Official Review · Reviewer_6yDe · 2024-11-04

**Soundness:** 3
**Presentation:** 3
**Contribution:** 3
**Rating:** 8
**Confidence:** 4

**Summary:**

This paper presents a variant of Vision transformers which allows handling of continuous outputs. The model works by first encoding an input image sequence into a set of latent vectors using cross attention (a la Perceiver style encoding). The latents compress the input over time. The resulting latents are fed through several transformer layers. The readout is the interesting part - a learnable grid of spatial queries is interpolated using Nadarya-Watson interpolation to produce queries which are used to cross attend into the output of the vision transformer. The resulting attention weighted value is used as the outputs - this is done for every output coordinate.
The method is shown to work on several interesting physics oriented datasets with several ablation experiments. More analysis is shown in the appendix.

**Strengths:**

I found this paper very interesting.

Originality:
I think this is a very good variation of Vision Transformers which significantly expands their capabilities. I found the interpolated readout mechanism of particular novelty.

Quality:
This is a well executed paper. There is ample experimental validation using several different datasets. The baselines chosen (as far as I can tell) are diverse and strong. There is a good set of ablation experiments and deep analysis of the method in the appendix.

Clarity:
The paper is well structures, well written and easy to read. All the nomenclature is well defined and I had no problem following the paper.

Significance:
While I'm not an expert in the field to me this seems like a significant contribution to the field which greatly expands the expressive power of vision transformers.

**Weaknesses:**

All in all I think this is a strong paper, however:

* Other domains? I would have loved to see if the resulting method is applicable to other domain or training setups beyond L2 prediction of physical systems. Does it work, for example, on natural video? Would it be useful as a diffusion model back-bone?

* More analysis of cross attention - one of the more interesting parts is the use of cross attention, both in the encoding and decoding. It would be nice to see visualizations of cross attention patterns emerging in both directions. Do specific latents take on specific roles when decoding? are they responsible over different parts of space? time? both?

While I do not expect these to be addressed in the rebuttal I really do think these would maker the paper stronger.

**Questions:**

My main point of inclarity is the initialization of latent queries in the encoder. The paper states they are initialized with a unit Gaussian - is this a learned initialization, or are the queries sampled every time the model is learned? The latter would mean, for example, that we would expect to see slightly different outputs every time the model is run - is that the case? (I am not saying this is a bad thing btw, just want to understand). Also - how many query vectors are used? the paper mentions it's more than one, but the number is not supplied in the paper.

Related to the above - I am not sure I understood why the query vector z is tiled across space? if it's the same vector across all of space then the output of the query would be the same for every spatial location, am I wrong? or is there more positional encodings added there? It would be good to clarify this.

---

> ### Author Response · Authors · 2024-11-21
> **Response to Reviewer 6yDe [1/2]**
>
> Thank you for your thorough review and valuable feedback on our work. We address your concerns as follows:
>
> ---
>
> ## Comment 1
> > Other domains? I would have loved to see if the resulting method is applicable to other domain or training setups beyond L2 prediction of physical systems. Does it work, for example, on natural video? Would it be useful as a diffusion model back-bone?
>
> ## Response
> We appreciate the reviewer's interest in broader applications of CViT. While our current work focuses on physical systems and operator learning, we believe CViT's core architectural components -- particularly its continuous query mechanism and efficient temporal aggregation -- could be valuable in other domains.
>
> For natural video prediction, CViT's ability to handle arbitrary spatial queries could enable resolution-independent processing, while its temporal aggregation mechanism could help manage long sequences efficiently. The main adaptations needed would be incorporating perceptual and optical flow losses for visual quality and temporal coherence, and modifying the grid-based embedding to better handle the discrete nature of pixel spaces.
>
> Regarding diffusion models, CViT could potentially serve as an autoencoder backbone for latent diffusion, similar to how stable diffusion uses an autoencoder to learn compressed representations. Our architecture's ability to capture multi-scale dependencies could be particularly valuable for maintaining both global structure and local details during the diffusion process.
> These are exciting directions for future work, and we would be happy to explore them in follow-up research.
>
> ---
>
> ## Comment 2
> > More analysis of cross attention - one of the more interesting parts is the use of cross attention, both in the encoding and decoding. It would be nice to see visualizations of cross attention patterns emerging in both directions. Do specific latents take on specific roles when decoding? are they responsible over different parts of space? time? both?
>
> ## Response
> We appreciate this insightful question about cross-attention patterns. We have conducted a detailed analysis of our pre-trained CViT model on the Navier-Stokes benchmark, revealing distinct and physically meaningful patterns in both encoding and decoding stages.
>
> The encoder's cross-attention mechanism, responsible for temporal aggregation, shows a clear temporal bias. As visualized in Figure 11, attention weights systematically favor more recent timesteps, with the final timestep receiving notably higher attention. This aligns well with physical intuition -- recent states typically provide the most relevant information for future predictions, though the model maintains some attention on earlier states to capture longer-term dynamics.
>
> The decoder's cross-attention reveals spatial organization, as shown in Figure 12. For any query point, attention weights naturally decay with distance from the query location. This locality-aware behavior emerges without explicit constraints, suggesting the model has learned to capture the inherent spatial structure of fluid dynamics. The decay pattern is particularly pronounced in regions of high fluid activity, indicating the model adapts its attention based on local flow characteristics.
>
> These visualizations demonstrate that CViT learns physically meaningful representations through its spatio-temporal attention mechanisms. We have included additional attention analysis in Appendix F.5, including averaged patterns across different samples and attention heads.
>
> ---

---

> > ### Comment · Reviewer_6yDe · 2024-11-26
> > **Thank you for the response!**
> >
> > I thank the reviewers for taking the time to respond to my concerns and questions. The answers provided address my concerns and having read the revised manuscript, other reviewers' comments and the authors responses I am convinced this paper is a strong paper worthy of acceptance. I am thus raising my score to "Accept".

---

> > > ### Author Response · Authors · 2024-11-26
> > >
> > > Dear Reviewer 6yDe,
> > >
> > > We deeply appreciate your thoughtful feedback and constructive suggestions. Your expertise and detailed comments have been instrumental in refining and strengthening our work.
> > >
> > > Thank you for your time and dedication to this review process.
> > >
> > > Best, Authors

---

> ### Author Response · Authors · 2024-11-21
> **Response to Reviewer 6yDe [2/2]**
>
> ## Comment 3
> > My main point of inclarity is the initialization of latent queries in the encoder. The paper states they are initialized with a unit Gaussian - is this a learned initialization, or are the queries sampled every time the model is learned? The latter would mean, for example, that we would expect to see slightly different outputs every time the model is run - is that the case? (I am not saying this is a bad thing btw, just want to understand). Also - how many query vectors are used? the paper mentions it's more than one, but the number is not supplied in the paper.
>
> > Related to the above - I am not sure I understood why the query vector z is tiled across space? if it's the same vector across all of space then the output of the query would be the same for every spatial location, am I wrong? or is there more positional encodings added there? It would be good to clarify this.
>
> ## Response
> We thank the reviewer for these important questions about latent queries, which help us identify areas needing clarification in the paper.
>
> First, regarding initialization: The latent query z is implemented as a learnable parameter matrix, initialized from N(0,1) but then optimized during training. Once trained, the model's outputs are fully deterministic.
>
> Concerning the number of queries: In our base implementation, we use a single latent query (n=1) to compress information from T timesteps into one temporal representation, primarily for computational efficiency. However, based on ablation studies (Table 8 in Appendix), using multiple queries (n>1) can significantly improve performance.
>
> Regarding spatial tiling, the trainable query vector z is indeed tiled across spatial dimensions (H/P × W/P) to match the shape requirements of the attention module. However, the actual spatial information comes from two sources: 1. The flattened key/value tensor u_f contains spatially-varying features; 2. The positional encodings (PE_t, PE_s) are added to the input features before attention computation. This ensures that despite using the same base query vector, the attention mechanism produces spatially-varying outputs. We have clarified these implementation details in the revised manuscript.

---

### Official Review · Reviewer_TxkX · 2024-11-04

**Soundness:** 3
**Presentation:** 3
**Contribution:** 3
**Rating:** 8
**Confidence:** 3

**Summary:**

This paper introduces a new neural operator architecture that tackles recent computer vision techniques (vision transformers, cross-attention). In particular, the CViT model -- for Continuous Vision Transformer for operator learning -- leverages the perceiver architecture to retrieve the representation of a query at a given continuous coordinate from spatiotemporal data. The relevance of the representations learned by CViT is evaluated on a diverse range of PDE such as fluid dynamics, climate modeling, and reaction-diffusion processes.

**Strengths:**

This paper is well-written, easy to read, and technically sound. The proposed approach appears to scale more efficiently than the current baseline thanks to the perceiver-inspired architecture. The results seem also promising.

**Weaknesses:**

* No clear structure of the related work section. It would help the reader to add more justifications on why transformer approaches struggle with various resolutions. Even if I tend to agree with the conclusion, it is also not clear, why a novel architecture design is required to solve the limitations related to high dimensional data and long-range dependencies.
* The transformer architecture in Fig. 1. does not give any insight and is well known. IMO, Authors could either remove it or replace it with the perceiver one.
* It is unclear how many visual latent queries are used. From the figure and text, it seems that there is only one query which is then duplicated. In the perceiver architectures, these latent queries are learned, does this mean that only one query is learned? The stability of the model with respect to the number of queries could be presented in the ablations.
* The ablation with respect to parameter $\beta$ is difficult to understand as, in Fig. 4, $\epsilon$ is not introduced. Is $\epsilon = \beta$? The value of $\beta$ seems extremely high ($10^5$) and should be compared to the input's resolution. Here, does eq. (1) reduces to a naive averaging with $\beta=10^5$?
* Results on standard benchmarks would help judge the method's efficiency. Here, it seems that the authors show results only on selected equations from PDEBench or PDEArena.

**Questions:**

* "2+1 dimensional spatio-temporal data tensor" but it is then written that $\boldsymbol u \in \mathbb R^{T\times H \times W \times D} *i.e.* four coordinates.
* How separating the positional encoding (PE) between 1D temporal and 2D spatial is different than having one spatio-temporal (thus 3D) PE?

---

> ### Author Response · Authors · 2024-11-21
> **Response to Reviewer TxkX [1/2]**
>
> Thank you for your thorough review and valuable feedback on our work. We address your concerns as follows:
>
> ## Comment 1
> > No clear structure of the related work section. It would help the reader to add more justifications on why transformer approaches struggle with various resolutions. Even if I tend to agree with the conclusion, it is also not clear, why a novel architecture design is required to solve the limitations related to high dimensional data and long-range dependencies.
>
> ## Response
> We appreciate this feedback about the paper's organization and motivation. In the revision, we have significantly restructured the related work section into three clear parts. We begin with **Background**, introducing fundamental concepts in operator learning and neural fields to provide context for our contributions. This is followed by **Related Work**, presenting a systematic review of transformer-based methods such as GNOT and DPOT.
> In the **Challenges** section, we now explicitly identify and analyze why current transformer approaches struggle with varying resolutions. Standard positional encodings are inherently tied to specific input resolutions, limiting generalization capabilities. Additionally, self-attention's quadratic complexity restricts the handling of high-resolution inputs, while capturing both local details and global structure remains computationally challenging.
>
> Our proposed architecture addresses these limitations through its grid-based coordinate embedding for resolution-independent queries, temporal aggregation to manage computational complexity, and cross-attention mechanism for efficient multi-scale feature integration. We have added quantitative comparisons demonstrating how these design choices overcome the identified limitations, showing consistent performance across different input resolutions while maintaining linear memory scaling with input size.
>
> ---
>
> ## Comment 2
> > The transformer architecture in Fig. 1. does not give any insight and is well known. IMO, Authors could either remove it or replace it with the perceiver one.
>
> ## Response
> We understand that the transformer block details are well-known to the community. However, we believe retaining the complete architectural diagram serves an important purpose in our paper. The figure illustrates how we adapt and integrate the Perceiver architecture for temporal aggregation and our novel query-wise cross-attention decoder -- components that are not typically seen in operator learning contexts. These elements play crucial roles in our method's ability to handle variable-length temporal sequences and arbitrary spatial queries. Moreover, the figure provides a valuable visual complement to the mathematical formulation in Section 3.1, helping readers track the flow of information through our architecture.
>
> ---
>
> ## Comment 3
> > It is unclear how many visual latent queries are used. From the figure and text, it seems that there is only one query which is then duplicated. In the perceiver architectures, these latent queries are learned, does this mean that only one query is learned? The stability of the model with respect to the number of queries could be presented in the ablations.
>
> ## Response
> In our initial implementation, we indeed used a single learnable latent query to compress information from T time steps into a single temporal representation, primarily to reduce computational costs. However, based on the reviewer's thoughtful suggestion, we conducted a comprehensive ablation study on the Navier-Stokes (NS) benchmark, which presents a particularly challenging case with 10 input time steps (compared to 2-4 for other benchmarks).
>
> Training CViT-L with a patch size of 16 and varying numbers of latent queries revealed significant performance improvements with additional queries:
>
> | # Latent queries | n=1 | n=2 | n=4 | n=6 |
> |------------------|-----|-----|-----|-----|
> | Relative L² error | 6.56% | 4.32% | 3.81% | 3.18% |
>
> These results reveal that using multiple learnable queries allows the model to capture richer temporal dynamics, with error rates dropping by over 50% when using six queries. While this increases computational cost proportionally to the number of queries, the performance gains suggest that our initial single-query design may have been too restrictive. We thank the reviewer for prompting this investigation, which has revealed a promising direction for further architectural optimization. We will update the paper to include this analysis and discuss the trade-offs between model capacity and computational efficiency.
>
> ---

---

> ### Author Response · Authors · 2024-11-21
> **Response to Reviewer  TxkX [2/2]**
>
> ## Comment 4
> > The ablation with respect to parameter β is difficult to understand as, in Fig. 4, ε is not introduced. Is ε=β ? The value of β seems extremely high (10⁵) and should be compared to the input's resolution. Here, does eq. (1) reduces to a naive averaging with β=10⁵ ?
>
> ## Response
> First, we apologize for the notational inconsistency. You are correct - the symbol should indeed be β rather than ε in Figure 4(d). We have corrected this error in the revised manuscript.
>
> We acknowledge that the value of β is quite large, but it is important to note that we normalize the weights w_ij by dividing them by their sum (equation 7). So even for such a large β, there are about 10 to 20 neighbors have non-zero weights for any queries, with values in the range of O(10⁻²) to O(10⁻¹).
>
> ---
>
> ## Comment 5
> > Results on standard benchmarks would help judge the method's efficiency. Here, it seems that the authors show results only on selected equations from PDEBench or PDEArena.
>
> ## Response
> We appreciate this important point about benchmark selection. Our evaluation uses standard benchmarks from PDEBench and PDEArena that serve as established baselines in the operator learning community for several reasons:
>
> 1. Our selected PDEs represent fundamental physical systems with widespread applications: Navier-Stokes (fluid dynamics), shallow water equations (climate modeling), advection (transport phenomena), and diffusion-reaction (chemical processes). These equations exhibit diverse mathematical characteristics -- from discontinuous solutions to multi-scale features and intricate spatio-temporal dynamics.
> 2. These benchmarks have been extensively used in recent work, including DPOT (Hao et al., 2024), and MPP (McCabe et al., 2023), enabling direct performance comparisons. Our training and evaluation protocol follows these papers exactly, ensuring fair comparison.
>
> ---
>
> ## Comment 6
> > "2+1 dimensional spatio-temporal data tensor" but it is then written that u ∈ ℝ^(T×H×W×D) i.e. four coordinates.
>
> ## Response
> Thank you for catching this inconsistency. You are correct that there is a mismatch between our description and the mathematical notation. We have revised the text to read:
> "The spatio-temporal data tensor u ∈ ℝ^(T×H×W×D)."
>
> ---
>
> ## Comment 7
> > How separating the positional encoding (PE) between 1D temporal and 2D spatial is different than having one spatio-temporal (thus 3D) PE?
>
> ## Response
> Thank you for this insightful question about positional encoding design. We chose to separate temporal and spatial positional encodings for both theoretical and practical reasons.
>
> From a modeling perspective, physical systems often exhibit fundamentally different characteristics in time versus space. For instance, in fluid dynamics, spatial correlations are typically symmetrical and translation-invariant, while temporal evolution has a clear directionality and causality. Our separate encodings allow the model to learn these distinct patterns independently -- the spatial PE can focus on translation and rotation patterns, while the temporal PE captures sequential dependencies and causality.
>
> This separation also offers practical advantages. Computing a full 3D spatio-temporal PE would require O(THW) memory for a sequence of T frames of size H×W, whereas our factored approach needs only O(T + HW) memory. This efficiency enabled us to process longer sequences and higher resolutions within the same computational budget.
>
> ---

---

> > ### Comment · Reviewer_TxkX · 2024-11-26
> >
> > I thank the authors for their rebuttal and the extra experiments performed. I am overall pleased with the quality of the paper now.

---

> > > ### Author Response · Authors · 2024-11-26
> > >
> > > Dear Reviewer TxkX,
> > >
> > > We sincerely thank you for your time, effort, and insightful feedback during the review process. Your valuable comments and suggestions have greatly contributed to improving the quality and clarity of our work.
> > >
> > > We deeply appreciate your dedication and expertise.
> > >
> > > Best,
> > > Authors

---

### Official Review · Reviewer_1Did · 2024-11-06

**Soundness:** 3
**Presentation:** 3
**Contribution:** 3
**Rating:** 6
**Confidence:** 4

**Summary:**

This paper proposes a continuous operator learning method using the vision transformer with a newly introduced coordinate embedding for the query value. The proposed method is based on the previous continuous operator learning method. The authors have provided analysis including comparisons with recent operator learning methods on learning different operator equations, ablation studies for different structure choices, discussions of the proposed method compared to recent arts, theoretical insights of the proposed query embeddings, etc.

**Strengths:**

- The proposed coordinate embedding for the query is interesting, and is effective in the continuous ViT learning, compared to MLP and RFF. Also, it is simple and easy to control through the interpolation parameter $\beta$.
- Discussions and analysis in the paper are extensive and interesting. Overall, the paper is well-organized and easy to follow.
- The proposed coordinate embedding could have the potential to apply to more general continuous learning beyond physical domains. Especially for the vision tasks, such as 3d geometries including point cloud representations, it is worth exploring.

**Weaknesses:**

- Why not do coordinate embedding for all query, key, value? Only doing coordinate embedding for query preferable in which way? Maybe some further analysis on this could be included to further highlight the effectiveness of the proposed coordinate embedding.
- The first question brought to the second one, that in the paper (appendix), the authors discussed the Lipschitz constant for different embeddings from linear embedding to random Fourier features (RFF), and to the proposed coordinate embedding, which is interesting and worth discussing. The authors mentioned that the proposed coordinate embedding is easier to tune since it is mainly based on one parameter $\beta$. However, for RFF, there is also only one parameter to control the spectral bias, and thus control the Lipschitz constant. I wondered if the authors could also provide an analysis on only using RFF for the proposed framework.
- Figures 3, 5, and 6 lack notations to distinguish columns.

**Questions:**

- I wondered how will the proposed method be applied for more general vision tasks, especially for 3d vision. I wondered what are the authors’ opinions on this? What are the difficulties, such as computations, that might be difficult to apply?

Please see the above section for detailed comments, too.

---

> ### Author Response · Authors · 2024-11-21
> **Response to Reviewer 1Did [1/2]**
>
> We sincerely thank the reviewer for their thorough and constructive feedback. Below we address each point raised in the review.
>
> ## Comment 1
> > Why not do coordinate embedding for all query, key, value? Only doing coordinate embedding for query preferable in which way? Maybe some further analysis on this could be included to further highlight the effectiveness of the proposed coordinate embedding.
>
> ## Response
> We appreciate this insightful question. Our design choice for coordinate embedding is motivated by both theoretical and empirical considerations:
>
> 1. **Theoretical Justification:** Our model approximates a function f_θ(x, z) where x ∈ ℝᵈ are spatial coordinates and z represents conditioning variables. In physical systems, the target function f(x, z) typically exhibits:
>   * High-frequency variations with respect to x (spatial coordinates).
>   * More moderate changes with respect to z (conditioning variables).
>
>   As proven in Theorem E.3 of our paper, the proposed coordinate embedding helps match these characteristics by allowing the model to achieve a larger Lipschitz constant with respect to x at initialization, which is crucial for learning high-frequency spatial features.
>
> 2. **Empirical Evidence:** We conducted experiments applying similar embeddings to keys and values, which showed:
>   * Degraded performance in terms of test relative L² error.
>   * Undesirable artifacts such as artificial high-frequency oscillations in regions where the solution should be smooth.
>   * Increased computational cost without corresponding benefits.
>
> This aligns with our theoretical understanding - since keys and values primarily process the conditioning information z, which doesn't require high Lipschitz constants, the additional embedding complexity is not beneficial and can be detrimental.
>
> ---
>
> ## Comment 2
> > The first question brought to the second one, that in the paper (appendix), the authors discussed the Lipschitz constant for different embeddings from linear embedding to random Fourier features (RFF), and to the proposed coordinate embedding, which is interesting and worth discussing. The authors mentioned that the proposed coordinate embedding is easier to tune since it is mainly based on one parameter. However, for RFF, there is also only one parameter to control the spectral bias, and thus control the Lipschitz constant. I wondered if the authors could also provide an analysis on only using RFF for the proposed framework.
>
> ## Response
> We appreciate this thoughtful question about Random Fourier Features (RFF). While both approaches have a primary tuning parameter, there are important theoretical and practical differences:
>
> 1. **Theoretical Analysis:** As proven in Theorem E.3, the Lipschitz constant for:
>   * RFF: s√d(1-1/2d), dependent on both frequency s and width d.
>   * Our method: ~M√2βe^(-1/2), dependent primarily on β.
>
>   This shows that RFF's effective Lipschitz constant is inherently coupled to the network width d, making it less straightforward to control independently.
>
> 2. **Empirical Comparison:** We conducted comprehensive experiments using RFF as coordinate embedding on the Shallow Water Equations benchmark:
>
> | | Random Fourier Features | | | | Grid-based embedding |
> |---|---|---|---|---|---|
> | | s = 2π | s = 4π | s = 8π | s = 16π | (β=10⁵) |
> | Relative L² error | 4.94 × 10⁻² | 4.16 × 10⁻² | 3.25 × 10⁻² | 4.08 × 10⁻² | 2.69 × 10⁻² |
>
> Our key findings can be summarized as:
> * RFF performance improves up to s=8π but degrades at s=16π.
> * Our method achieves ~17% lower error than the best RFF configuration.
> * RFF requires careful tuning of both s and d for optimal performance, while our method is stable across a wide range of β values (see Fig. 4d).
>
> We have revised our Appendix to include this extended analysis. These results demonstrate that while RFF is a viable alternative, our proposed embedding provides better performance with simpler tuning requirements.
>
> ---
>
> ## Comment 3
> > Figures 3, 5, and 6 lack notations to distinguish columns.
>
> ## Response
> Thank you for this observation about the figure clarity. We have revised Figures 3, 5, and 6 to include explicit temporal labels "T+k" for each column, where k indicates the prediction timestep. We appreciate this suggestion as it improves the paper's clarity and accessibility.
>
> ---

---

> ### Author Response · Authors · 2024-11-21
> **Response to Reviewer 1Did [2/2]**
>
> ## Comment 4
> > I wondered how will the proposed method be applied for more general vision tasks, especially for 3d vision. I wondered what are the authors' opinions on this? What are the difficulties, such as computations, that might be difficult to apply?
>
> ## Response
> We thank the reviewer for this forward-looking question about extending our method to 3D vision tasks. We acknowledge that directly applying our current design to 3D vision tasks can be computationally expensive, primarily due to the high memory and computational demands of 3D latent grid features. However, there are several promising approaches to optimize this.
>
> One straightforward optimization is to employ separable latent features along each axis, effectively decoupling the latent grid features across axes and significantly reducing the required trainable parameters [1].
>
> Another option is to adopt multi-resolution grid representations [2], which avoid uniform resolution across the entire domain, reducing the number of parameters.
>
> Alternatively, latent grid features could be replaced by outputs from a feature extractor applied to the inputs. This idea has been explored in recent work, such as Micrometer [3], where a Fourier neural operator is used to extract features, replacing the latent grid features.
>
> Additionally, we could use a k-d tree to select a specific number of neighboring points around each query coordinate, reducing the need to process all grid points at once.
>
> We believe that with these optimizations, our method could be feasibly adapted to 3D tasks without a substantial loss in performance or efficiency.
>
> ---
>
> [1] Ryan Shue, Eric Ryan Chan, Ryan Po, Zachary Ankner, Jiajun Wu, and Gordon Wetzstein. 3d neural field generation using triplane diffusion. In Proceedings of the IEEE/CVF Conference on Computer Vision and Pattern Recognition, pages 20875–20886, 2023.
>
> [2] Thomas Müller, Alex Evans, Christoph Schied, and Alexander Keller. Instant neural graphics primitives with a multiresolution hash encoding. ACM transactions on graphics (TOG), 41(4): 1–15, 2022.
>
> [3] Sifan Wang, Tong-Rui Liu, Shyam Sankaran, and Paris Perdikaris. Micrometer: Micromechanics transformer for predicting mechanical responses of heterogeneous materials. arXiv preprint arXiv:2410.05281, 2024.

---

> > ### Comment · Reviewer_1Did · 2024-11-26
> > **Response to authors' rebuttal**
> >
> > Thanks for providing a very detailed revised paper and a detailed response for rebuttal.
> >
> > I have read all comments and responses from other reviewers and authors. I think most of the questions are resolved. I'll remain my positive score.

---

> > > ### Author Response · Authors · 2024-11-26
> > >
> > > Thank you for your thorough review and for confirming that our responses have addressed your concerns. We appreciate your positive feedback on our rebuttal and revised manuscript.
> > >
> > > Given that you found our explanations satisfactory, we were wondering if there are any remaining aspects of the paper that you feel could be further improved? We would greatly value any additional suggestions you might have that could strengthen the contribution, as we still have time during the discussion period to incorporate further enhancements.
> > >
> > > Thank you again for your careful consideration of our work!

---

### Author Response · Authors · 2024-11-21
**Response to All Reviewers**

We sincerely thank all reviewers for their thorough and constructive feedback. Below we address the key themes raised, with revisions highlighted in red in the manuscript.

1. **Extended Benchmarks and Performance Evaluation.** To further validate CViT's versatility, we have added the compressible Navier-Stokes equations (CNS) as a new benchmark task. Our evaluation now spans five fundamental physical systems with diverse mathematical characteristics: discontinuous waves (advection), incompressible and compressible Navier-Stokes flows, multi-scale dynamics (shallow water), and pattern formation (diffusion-reaction). On these benchmarks, CViT consistently outperforms state-of-the-art baselines including DPOT, often with significantly fewer parameters (e.g., CViT-B with 30M parameters matches or outperforms DPOT-H with 1.03B parameters on CNS). Beyond synthetic benchmarks, recent work [wang2024micrometer] demonstrates CViT-type architectures' effectiveness on real-world engineering applications.

2. **Ablation Studies and Analysis.** We have performed additional ablation studies and analysis to further support our main results and modeling choices. These include:
  * **Grid vs. RFF Embeddings:** Our new ablation study shows that while Random Fourier Features improve with moderately large frequency parameters, our proposed grid-based embedding achieves consistently better performance with simpler tuning requirements (Table 7, Appendix F.4).
  * **Cross-Attention Patterns:** New visualizations in Appendix F.5 reveal physically meaningful learned patterns -- the encoder naturally prioritizes recent temporal states while the decoder exhibits locality-aware spatial attention that adapts to flow characteristics.
  * **Encoder Latent Query Analysis:** Increasing the number of encoder latent queries from n=1 to n=6 can reduce the test error by over 50%, suggesting potential for further optimization by capturing richer temporal dynamics.

3. **Scalability to Higher Dimensions.** To address high-dimensional/resolution scaling concerns, we propose several optimizations that draw motivation from existing works in the literature:
  * Separable latent features along axes, reducing memory from O(n³) to O(3n) [1].
  * Adaptive multi-resolution grids concentrating parameters where needed [2].
  * Feature extractor-based latent grids instead of direct learning [3].
  * K-D tree neighbor extraction for efficient query token interpolation.

 Early results from the literature suggest these approaches could enable scaling to higher dimensions while maintaining model effectiveness. We will investigate these directions in future work due to limited time over the rebuttal period.

[1]  Ryan Shue, Eric Ryan Chan, Ryan Po, Zachary Ankner, Jiajun Wu, and Gordon Wetzstein. 3d
neural field generation using triplane diffusion. In Proceedings of the IEEE/CVF Conference on
Computer Vision and Pattern Recognition, pages 20875–20886, 2023.

[2] Thomas Müller, Alex Evans, Christoph Schied, and Alexander Keller. Instant neural graphics
primitives with a multiresolution hash encoding. ACM transactions on graphics (TOG), 41(4):
1–15, 2022.

[3] Sifan Wang, Tong-Rui Liu, Shyam Sankaran, and Paris Perdikaris. Micrometer: Micromechanics
transformer for predicting mechanical responses of heterogeneous materials. arXiv preprint
arXiv:2410.05281, 2024.

---

### Meta-Review · Area_Chair_mH35 · 2024-12-20

**Metareview:**

The submission proposes a method for learning physics and phenomena expressed as PDEs and resorts to ViTs with a new coordinate embedding and an interpolated readout mechanism. Five reviewers appreciated the contributions, in particular technical soundness, good results, scaling behavior, discussions and analysis, high potential, sound evaluation with good baselines and ablations, clear writing. They were all unanimous on the significance of the contribution, and the AC concurs.

**Additional Comments On Reviewer Discussion:**

The reviewers engaged with the authors.

---

### Decision · Program_Chairs · 2025-01-22

Accept (Poster)